# Pronounced centennial-scale Atlantic Ocean climate variability correlated with Western Hemisphere hydroclimate

Kaustubh Thirumalai [1,2,5], Terrence M. Quinn[1,2], Yuko Okumura[1], Julie N. Richey[3], Judson W. Partin[1], Richard Z. Poore[3] & Eduardo Moreno-Chamarro[4]

Surface-ocean circulation in the northern Atlantic Ocean influences Northern Hemisphere climate. Century-scale circulation variability in the Atlantic Ocean, however, is poorly constrained due to insufficiently-resolved paleoceanographic records. Here we present a replicated reconstruction of sea-surface temperature and salinity from a site sensitive to North Atlantic circulation in the Gulf of Mexico which reveals pronounced centennial-scale variability over the late Holocene. We find significant correlations on these timescales between salinity changes in the Atlantic, a diagnostic parameter of circulation, and widespread precipitation anomalies using three approaches: multiproxy synthesis, observational datasets, and a transient simulation. Our results demonstrate links between centennial changes in northern Atlantic surface-circulation and hydroclimate changes in the adjacent continents over the late Holocene. Notably, our findings reveal that weakened surface-circulation in the Atlantic Ocean was concomitant with well-documented rainfall anomalies in the Western Hemisphere during the Little Ice Age.

[1] Institute for Geophysics, Jackson School of Geosciences, University of Texas at Austin, J.J. Pickle Research Campus, Building 196, 10100 Burnet Road (R2200), Austin, TX 78758, USA. [2] Department of Geological Sciences, Jackson School of Geosciences, University of Texas at Austin, 1 University Station C9000, Austin, TX 78712, USA. [3] U.S. Geological Survey, St. Petersburg Coastal and Marine Science Center, 600, Fourth Street South, St. Petersburg, FL 33701, USA. [4] Department of Earth, Atmospheric and Planetary Sciences, Massachusetts Institute of Technology, Cambridge, MA 02139, USA. [5] Present address: Department of Earth, Environmental and Planetary Sciences, Brown University, 324 Brook Street, Providence, RI 02912, USA. Correspondence and requests for materials should be addressed to K.T. (email: kaustubh_thirumalai@brown.edu)

Surface circulation in the Atlantic Ocean is an integral process to the global climate system as it facilitates heat transport between the hemispheres[1]. For instance, the Gulf Stream, a surface current associated with the Atlantic Meridional Overturning Circulation (AMOC), transports heat and salt poleward into the North Atlantic, and influence North American and European climate[1,2]. However, variability in Atlantic surface circulation, including the Gulf Stream and other components, can be driven by processes outside AMOC variability, including wind-driven changes and regional gyre circulation, and still, have considerable impacts on climate variability[3,4]. Due to the short length of the instrumental record, the extent to which surface-ocean circulation in the Atlantic responds to atmospheric forcing or internal ocean dynamics versus changes in meridional over-turning, on the timescales of centuries is yet unresolved[2,5–8]. Furthermore, the implications of such century-scale fluctuations in surface-ocean circulation for terrestrial hydroclimate variability are also not well-known.

Previous studies have attempted to document variability in Atlantic Ocean circulation and its link with climate variability over the past millennium, but the timing, magnitude, impacts, and spatial expression of these changes remain uncertain. Reconstructions of Florida Current transport (purple stars in Fig. 1;[9], a precursor current that directly feeds into the Gulf Stream, suggest a ~10% decrease in transport volume during the Little Ice Age (LIA; defined hereinafter as 1450–1850 C. E. based on the IPCC[10]). Northward of the path of the Gulf Stream, a radiocarbon record of water mass variability from the North Iceland shelf (purple star in Fig. 1), interpreted to record variations in the North Atlantic Current, corroborates this reduction in Gulf Stream transport during the LIA[11]. However, the extent to which these reductions in surface-circulation and transport are coupled to reductions in AMOC variability is under debate[12], with one recent study[13] finding equivocal evidence for AMOC reductions during the LIA. Regardless of the link with AMOC variability, if the reconstructions of the Florida Current transport and the North Atlantic current are accurate, a reduction of Atlantic Ocean surface circulation during the LIA ought to be evident in ocean circulation systems both upstream and down-stream of the Gulf Stream[1,2].

In this regard, the Loop Current is situated upstream of the Gulf Stream and feeds warm, salty Caribbean waters into the Florida Current via the Yucatan Channel[14]. Through eddy-shedding processes, the Loop Current also transports water masses into the Gulf of Mexico (hereafter GOM) and affects the sea-surface temperature (SST), salinity (SSS), and circulation of the region[14,15]. However, no records exist that directly capture the strength of the Loop Current over the LIA.

Here we address this shortfall and reconstruct SST and SSS variability over the last 4,400 years using foraminiferal geochemistry in marine sediments cored from the Garrison Basin (26°40.19′N,93°55.22′W, blue circle with yellow border in Fig. 1), northern GOM. We make inferences about past changes in Loop Current strength by identifying time periods in our reconstruction where synchronous decreases in SST and SSS are interpreted as periods with a weaker Loop Current due to reduced eddy penetration over that period and vice versa[14,15]. Thus, we assess the spatial heterogeneity of the putative reduction of Atlantic surface-ocean circulation and furthermore, with multiproxy synthesis, correlation analysis, and model-data comparison, we document linkages between changes in Atlantic surface-circulation and Western Hemisphere hydroclimate anomalies. Our findings reveal that regardless of whether changes in the AMOC and deepwater formation occurred or not, weakened surface-circulation prevailed in the northern Atlantic basin during the Little Ice Age and was concomitant with widespread and well-documented precipitation anomalies over the adjacent continents.

## Results

**Proxy approach and uncertainty.** We measured magnesium-to-calcium ratios (Mg/Ca) and the stable oxygen isotope composition ($\delta^{18}O_c$) in the calcite tests of planktic foraminifer *Globigerinoides ruber* (white variety; *G. ruber* (W) hereafter) from multicores in the Garrison Basin, GOM (Fig. 1) as proxies for SST and SSS variability. Sediment trap results from a mooring close to the coring site demonstrate that *G. ruber* (W) is a suitable specimen for reconstructing mean annual sea-surface conditions from its geochemistry[16]. To assess the fidelity and preservation of

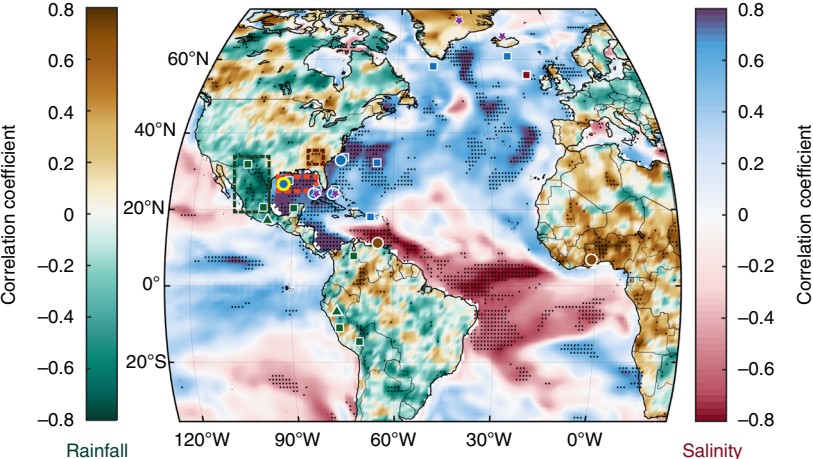

**Fig. 1** Long-term correlations between observed sea-surface salinity and rainfall. Correlation map between northern Gulf of Mexico sea-surface salinity (SSS; dashed red box) and global oceanic SSS (ORA-S4 data set; red-blue scale), as well as continental precipitation (GPCC data set; brown-green scale) with locations of proxy records used in the study. Proxy locations are marked with circles (sedimentary records), triangles (speleothems), dashed boxes (tree-ring compilations), stars (circulation proxies), and squares (additional proxies) with color fill indicating sign (fresh – blue; dry/wet–brown/green; purple–weakened poleward transport) during the Little Ice Age (1450–1850 C. E.). Correlations were performed with 8-year lowpass filters to reduce sensitivity to interannual variability, where black stippling indicates significance at the 5% confidence level

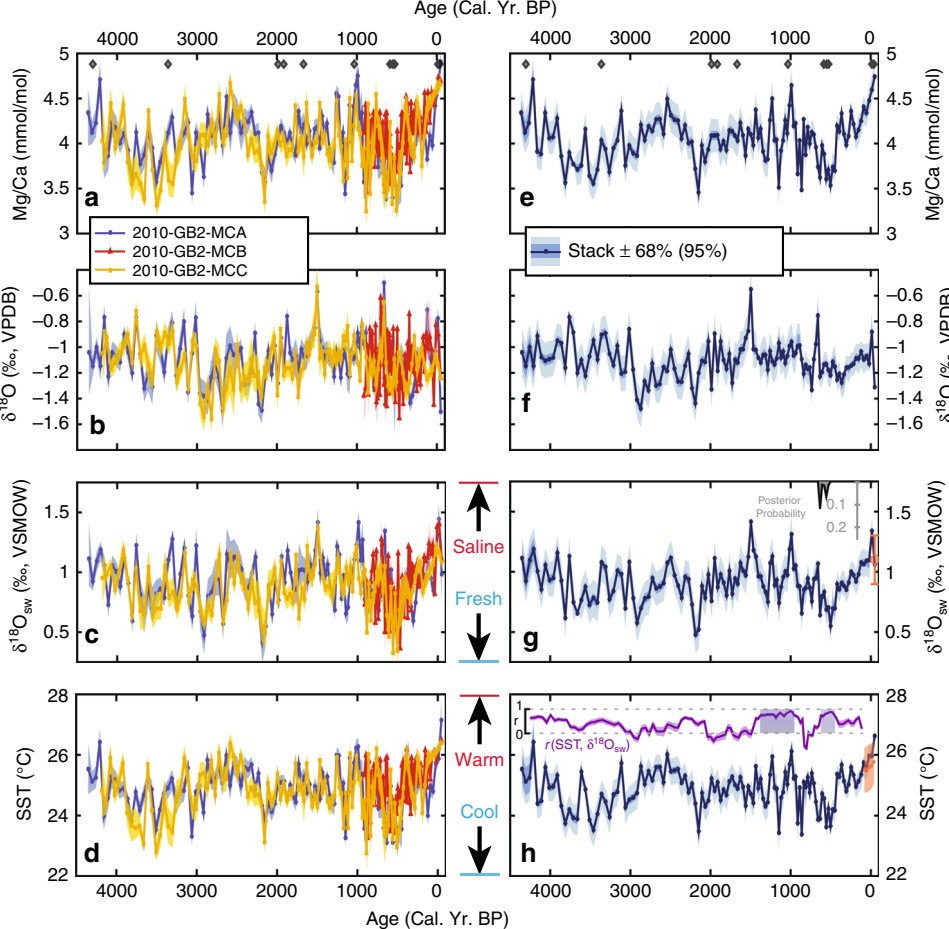

**Fig. 2** Garrison Basin multicore reconstructions and corresponding stacked records. Individual core Mg/Ca (mmol/mol) and $\delta^{18}O_c$ data (‰, VPDB), and $\delta^{18}O_{sw}$ (‰, VSMOW) and SST (°C) reconstructions (blue–MCA, red- MCB, yellow–MCC) plotted with median and 68% uncertainty envelope incorporating age, analytical, calibration, and sampling errors (**a-d**) along with corresponding median stacked records with 68% and 95% confidence bounds (**e-h**). Diamonds in **a** and **e** indicate stratigraphic points sampled for radiocarbon. Gray histogram in **g** is the probability distribution for a changepoint in the $\delta^{18}O_{sw}$ time series. Orange circle in **g** is the mean of available $\delta^{18}O_{sw}$ measurements in the GOM and orange line in **h** is observed monthly mean SST with uncertainty envelope calculated using a Monte Carlo procedure that simulates foraminiferal sampling protocol. Purple line in **h** is the 100-year running correlation between SST and $\delta^{18}O_{sw}$ with corresponding uncertainty with shaded boxes indicating correlations with $r > 0.7$ ($p < 0.001$), which is the basis for identifying time periods where Loop Current and associated processes are relevant

these proxy signals in the geological record, we performed paired Mg/Ca-$\delta^{18}O_c$ analyses on three multicores from the Garrison Basin (2010-GB2-MCA, 2010-GB2-MCB, 2010-GB2-MCC; hereafter MCA/MCB/MCC). Radiocarbon measurements on upper-ocean planktic foraminiferal tests (Supplementary Figure 1) revealed sedimentation rates of 14–16 cm kyr⁻¹ allowing for ~30-year sampling resolution (See Methods for details). Each multicore was independently dated and independently sampled for $\delta^{18}O_c$ and Mg/Ca measurements ($n = 70–100$) as a true test of replication on these high-resolution timescales. An age model was constructed using a bootstrap Monte Carlo procedure that incorporated the Bayesian posterior distributions of calibrated radiocarbon ages[17] from the multicores. We developed weighted, stacked records from the suite of multicore records with confidence levels accounting for analytical, sampling, and calibration errors for the $\delta^{18}O_c$ and Mg/Ca measurements[18,19], as well as replication errors (Fig. 2 and Supplementary Figure 2). Using a newly published algorithm, PSU Solver[19], we simultaneously calculated both SST and the $\delta^{18}O$ composition of the seawater ($\delta^{18}O_{sw}$), a proxy for SSS[9,20,21], over the length of the stacked records along with their uncertainty. We stress that structural uncertainties, such as the choice of calibration equations,

stationarity of the $\delta^{18}O_{sw}$-SSS relationship, and changes in calcification temperature and habitat might still affect the stacked reconstruction, although, the proximal sediment trap mooring provides confidence to downcore interpretation[16].

**Replication and validation.** The three Garrison Basin multicores display excellent replication in SST and $\delta^{18}O_{sw}$ (Fig. 2) wherein the confidence estimates for the corresponding stacked records provide for a large signal relative to the uncertainty. Replication minimizes dating, analytical, preservation, sampling, ecological, and calibration uncertainties, which become significant sources of interpretation bias in time periods with small signal-to-noise ratios, such as the late Holocene. To validate our SST reconstruction, we simulated how *G. ruber* (W) tests in a sediment core would record SST variability at the site using bootstrap Monte Carlo simulations[18] using the HadISST[22] data set (including analytical and sampling uncertainty). This exercise found a highly skillful match between instrumental SSTs and our reconstruction, where overlap is available (orange points and line in Fig. 2h; see Methods for more details). In addition, available measurements of $\delta^{18}O_{sw}$ in the northern GOM compared with $\delta^{18}O_{sw}$ computed at

the core-top also validate our downcore $\delta^{18}O_{sw}$ results (orange datapoint in Fig. 2g; Methods) and suggest that both stacked reconstructions are robust.

We note that recently compiled datasets of SSS[23] and $\delta^{18}O_{sw}$ observations[24] in the Atlantic (although these do not include data points from the GOM) indicate that advection of water masses and ocean circulation are a major influence on long-term variability in these parameters. EOF analysis on the newly compiled SSS observations from 1896 to 2013 in the tropical Atlantic indicates a century-long salinification trend[23], which is in line with the reconstructed $\delta^{18}O_{sw}$ record from the Garrison Basin multicores, although the temporal resolution of these records (as well as the synthesized records) and overall uncertainty precludes a quantitative comparison.

**SST and $\delta^{18}O_{sw}$ variability in Garrison Basin.** The Garrison Basin stacked records reveal marked centennial-scale variability in SST and $\delta^{18}O_{sw}$ (up to 1 °C and 0.25‰ respectively based on the running standard deviation of ~200-year bins) throughout the late Holocene (Fig. 2). We performed centennial running correlations (purple line in Fig. 2h, see Methods section for details) between the stacked reconstructions to track the co-variability of SST and $\delta^{18}O_{sw}$ to identify periods wherein Loop Current fluctuations might have occurred: synchronous, positively correlated changes are interpreted to be time periods with changes related to advection of Loop Current water masses[15]. This yielded two distinct periods having high positive correlations ($r > 0.75$, $p < 0.001$; purple shaded boxes in Fig. 2h): [1] ~650–1100 C. E., characterized by rapid and highly variable centennial-scale shifts in SST and $\delta^{18}O_{sw}$, and [2] ~1450–1650 C. E., during the initial part of the canonically defined LIA[10]. While the SST stack indicates cooler conditions in the Garrison Basin over the LIA, the $\delta^{18}O_{sw}$ stack, a proxy for SSS, shows a reduction in salinity that begins around ~1350 C. E., is at its freshest at ~1450 C. E. and displays an increasing trend into the current era. This event, situated at the onset of the LIA, is also identified as a time period containing a statistically significant changepoint (gray histogram in Fig. 2g) using a Bayesian methodology[25] in the stacked $\delta^{18}O_{sw}$ reconstruction. Over the full duration of the canonically defined LIA, the Garrison Basin record indicates considerably cooler (1 ± 0.2 °C) and fresher (0.23 ± 0.06‰; equivalent to a 0.5 ± 0.2 PSU reduction in salinity) conditions compared to the modern era (post-1850 C. E.)

**Loop Current control on regional SST and SSS variability.** We analyzed long-term (~multidecadal) observations in instrumental datasets to place our reconstructions into a global climatic context. The HadISST data set[22] documents 0.4–0.7 °C of multidecadal SST variability in the northern GOM over the last century (Fig. 3b). On these multidecadal timescales, SSTs in the northern GOM correlate highly with SST in the Loop Current region (Supplementary Figure 3). In particular, long-term SST variability here is impacted by the Loop Current through its eddy shedding processes[14] which are coupled to the strength of transport from the Yucatan Straits through the Florida Straits: if Loop Current transport is anomalously low, then northern GOM SSTs are anomalously cooler due to decreased eddy penetration and the opposite is the case when Loop Current transport is anomalously higher, i.e., northern GOM experiences anomalously warmer conditions[14,15,21]. Furthermore, the Loop Current, sitting upstream of where the Gulf Stream originates, correlates highly with SST associated with regions encompassing downstream currents (Supplementary Figure 3).

To track the spatial covariance of long-term SSS variations in the Garrison Basin, we correlated SSS datasets[26] in the northern

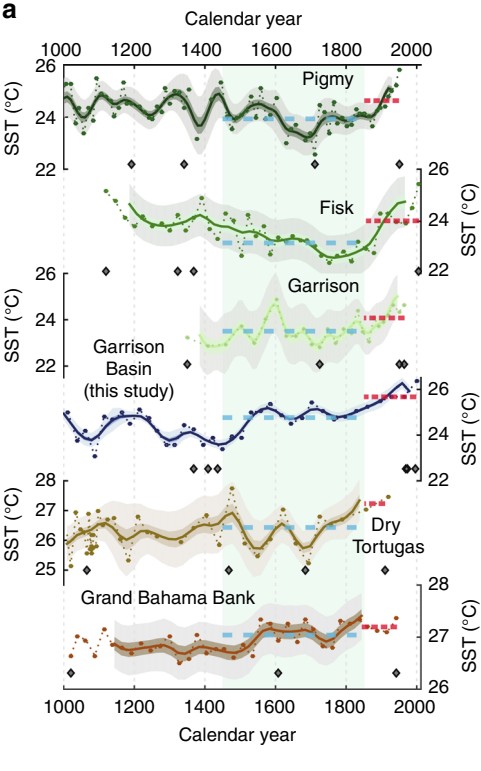

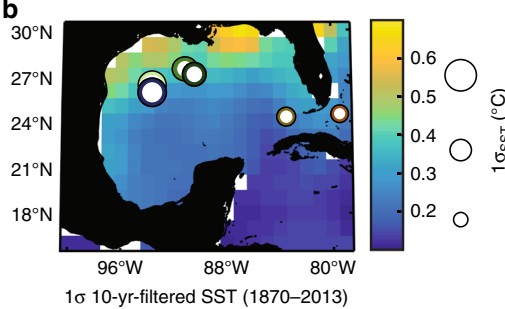

**Fig. 3** Regional comparison of SST records over the last millennium and proxy locations from the Gulf of Mexico. **a** Reconstructed SST data (circles connected by dotted lines) and 100-year filtered data (thick lines) with 68%, 95% confidence bounds plotted for each record. Previously published records are all reprocessed using the same calibration and uncertainty calculations as for the Garrison Basin records. Light green box highlights the Little Ice Age (1450–1850 C. E.) and blue and red dashed lines indicate the mean SST during the respective intervals. **b** Standard deviations of 10-year filtered SST from the monthly HadISST data set with proxy records overlain as white circles with border colors corresponding to above time series, scaled to indicate 1σ variability over the last millennium. Note that proxy SST variability is in line with expectations from instrumental SST variability (northern GOM 1σ-SST is larger than Florida Straits 1σ-SST)

GOM (red dashed box in Fig. 1) with SSS across the Atlantic Ocean on multidecadal timescales (Fig. 1). The resultant pattern yields a dipolar structure with significant correlation coefficients ($|r| > 0.7$; $p < 0.01$ − significance accounts for autocorrelation[27]) wherein GOM SSS correlates positively with the subtropical North Atlantic and western Caribbean Sea (blue patch), and correlates negatively with SSS along the northern South American coastline and tropical Atlantic Ocean (red patch). These SSS correlations cannot be explained by changes in evaporation-minus-precipitation alone across the Atlantic Basin, and along with SST changes, are suggestive of altered oceanic

currents as primary drivers (Methods). Essentially, large-scale advection tied to meridional surface-ocean transport in the Atlantic dictates these SSS correlation patterns on multidecadal timescales. Moreover, analyses on recently compiled observations of long-term SSS variability also confirm the importance of water mass advection as a major control on multidecadal timescales[23]. In summary, correlation analysis using SSS datasets provides a blueprint for investigating circulation variability and transport into the North Atlantic Ocean[28].

We also examine long-term correlations between SSS in the northern GOM and mean annual rainfall in the continents adjacent to the Atlantic Basin using rain-gauge precipitation datasets[29] (Fig. 1). Most notably, GOM SSS is anticorrelated with southern North American rainfall (i.e., fresher GOM with wetter southern North America) and is positively correlated with rainfall in West Africa, northern South America, and the southeast United States ($|r| > 0.6$, $p < 0.01$). These inferences demonstrate a correspondence between Western Hemisphere hydroclimate and Atlantic Ocean circulation on multidecadal timescales.

**Approach to understanding past circulation and hydroclimate**. Taken together, we interpret past periods in the Garrison Basin reconstructions when both SST and $\delta^{18}O_{sw}$ variability were positively correlated (salty/warm or fresh/cool) as periods during which Loop Current strength fluctuated. We hypothesize that during these periods, increased Loop Current penetration led to increased SST as well as increased advection of more enriched $\delta^{18}O_{sw}$ (or more saline waters) into the northern GOM. Using the correlation analysis as a blueprint[28], we can pinpoint whether these past fluctuations in the northern GOM $\delta^{18}O_{sw}$ record (such as during the LIA) were concomitant with changes in pan-Atlantic SSS records that would implicate circulation changes in the northern Atlantic Ocean. Finally, the long-term correlations with precipitation allow us to contextualize periods where surface-ocean circulation and continental rainfall anomalies were linked, which can then be placed within a multiproxy framework.

**Discussion**
We compiled published paleoceanographic records developed using *G. ruber* (W) in the GOM[30–32] and compared them with the new Garrison Basin record to investigate regional centennial-scale SST variability over the last millennium, and the LIA in particular (Fig. 3). We applied PSU Solver[19], our tool to model uncertainty, on the previously published records incorporating their age uncertainty, analytical and calibration errors, and numbers of foraminifera used for each record (see Methods for details). The northern GOM records (Garrison Basin − ours and a previous study[31], Fisk Basin[31], and Pigmy Basin[30]) all display higher SST variability (average $1\sigma_{last\ millennium} = 0.68 \pm 0.06\ °C$) than records from the Florida Straits (average $1\sigma_{last\ millennium} = 0.43 \pm 0.18\ °C$; Grand Bahama Bank & Dry Tortugas[32]). More variable northern GOM SST versus SST in the Florida Straits with lower variance is in line with expectations (Fig. 3b) from observed SST variability[22], boosting confidence in our proxy synthesis and uncertainty evaluation. As a whole, the GOM was between $0.6–1.1 \pm 0.7\ °C$ (at the 5% confidence level) cooler during the full interval of the canonically defined LIA time period compared to modern temperatures, with larger cooling in the northern GOM relative to the Florida Straits region[31,32]. We note the existence of discrepancies between these records over the entire duration of the LIA and note that the Garrison Basin stack indicates the onset of the LIA as an anomalous event compared to the overall record (Fig. 2). We infer this to probably reflect the inherent uncertainty in the previously published reconstructions, including limited

chronological constraints. Although, these differences might also arise from characteristic differences in multidecadal variability between the sites considering their proximity relative to Mississippi River discharge and the pathway of the Loop Current[33], highlighting the heterogeneous nature of SST variability in the GOM (Fig. 3b). Though the magnitude of our revised estimates of the cooler SSTs during the LIA period is less than previous studies[31] due to updated uncertainty constraints, *G. ruber* (W)-based SST reconstructions across different cores, basins, and different studies indicate a ~1 °C cooler GOM compared to modern temperatures.

Cooler SSTs in the GOM during the LIA are consistent with reduced Loop Current strength and reduced Gulf Stream volume transport[9]. A high-resolution ocean modeling study that investigated the influence of reduced Loop Current strength on GOM SSTs under a future warming scenario[14] supports this inference: weakened Loop Current transport has a cooling effect on GOM SST anomalies, which is particularly pronounced in the northern GOM, where the Garrison Basin is located. Although the LIA and current anthropogenic warming differ in their dynamical origins, the Garrison Basin SST stack and other GOM proxy records serve to extend our perspective on the effects of Loop Current and Atlantic circulation changes on regional variability and thereby allow insights into long-term climatic processes[34].

Following our regional SST synthesis, we compiled available paired Mg/Ca-$\delta^{18}O_c$ records of *G. ruber* (W) proximal to the Garrison Basin that resolved the LIA and computed $\delta^{18}O_{sw}$ with all aforementioned uncertainty considerations (Fig. 4). While the SST stack indicates cooler conditions in the Garrison Basin over the LIA compared to modern times, the $\delta^{18}O_{sw}$ stack, a proxy for SSS, shows a reduction in salinity as an event that begins around ~1350 C. E., is at its freshest at ~1450 C. E., and displays an increasing trend into the current era (Fig. 4a). The fresher conditions over the entire duration of the LIA (i.e., the difference in the 1450–1850 C. E. mean state compared to 1850 C. E. to present) is also observed in $\delta^{18}O_{sw}$-SSS records from the Dry Tortugas, Great Bahama Bank[32], and the Carolina slopes[35], all downstream of the GOM (blue circles with white boundaries in Fig. 1) considering the path[2] of the Gulf Stream (Fig. 4a–d). We note that akin to the comparison of SST records in Fig. 1, even though the Garrison Basin stack indicates an anomalous event at the onset of the canonically defined LIA, there are sub-centennial-scale discrepancies in the other $\delta^{18}O_{sw}$-SSS records which may arise due to methodological constraints, although, all of them display a multi-century trend towards more saline conditions from the LIA. The reprocessed $\delta^{18}O_{sw}$-SSS reconstructions appear to match the correlation map remarkably well (Fig. 1). The $\delta^{18}O_{sw}$ changes over the LIA in these records are also coeval with changes in seawater density reconstructions from the Florida Straits[9], and Greenland[36] ice core $\delta^{18}O$ where both records (purple stars in Fig. 1) provide potential evidence for a LIA reduction in Atlantic Ocean circulation (Fig. 4e–f), and potentially interhemispheric heat transport, although there are other factors to consider such as flow compensation[37] and ocean-atmosphere interactions[3]. Records of past SSS variability in addition to the aforementioned GOM and Carolina Banks sediment cores (Fig. 4) also match the correlation map. For example, SSS reconstructed from corals in Puerto Rico[38] and Bermuda[39] that extend into the 1700s, and paired Mg/Ca-$\delta^{18}O_c$ analysis on planktic foraminifera from North Atlantic sediment cores[40,41], also show lower $\delta^{18}O_{sw}$ during the LIA, indicating reduced salinities (blue squares with white boundaries in Fig. 1). Unfortunately, the lack of last millennium $\delta^{18}O_{sw}$ records from the tropical Atlantic precludes comparison with the correlation map. However, the proxy-observation match of anomalously fresh SSS

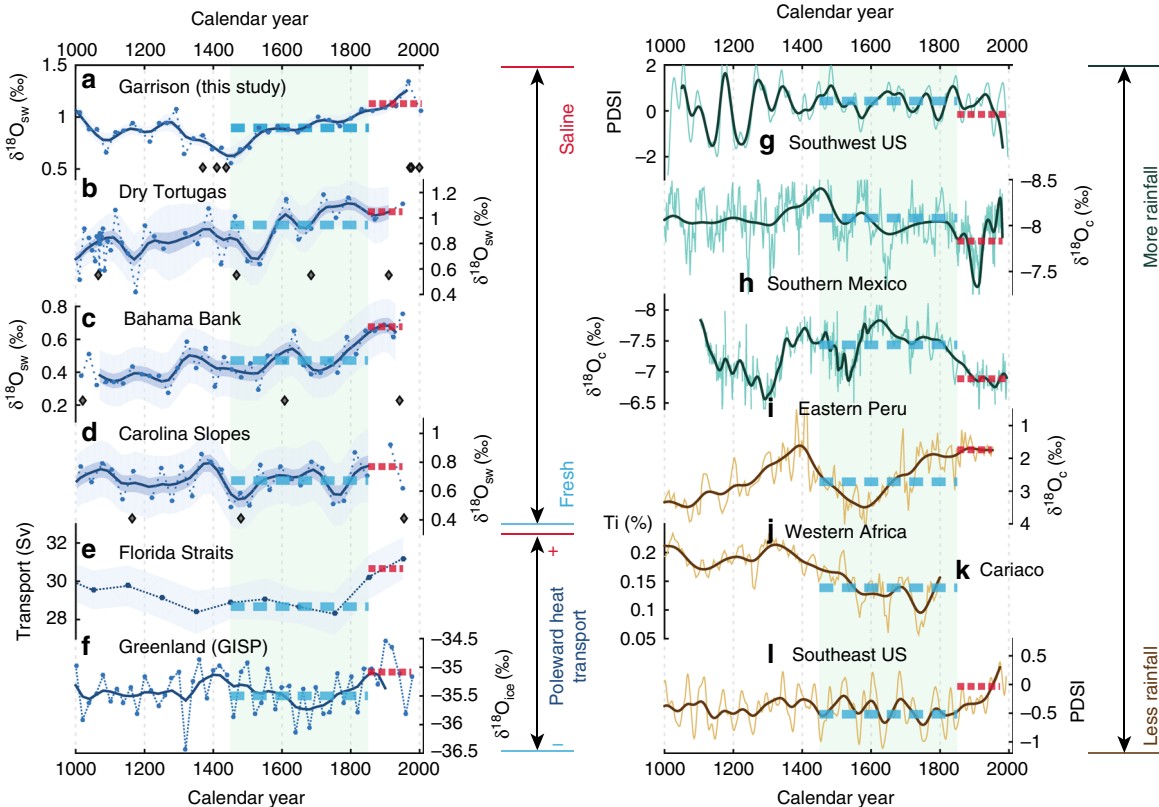

**Fig. 4** Comparison of SSS and precipitation proxies with poleward heat transport over the last millennium. Reconstructed $\delta^{18}O_{sw}$ data (circles connected by dotted lines) and 100-year filtered data (thick lines) with 68%, 95% confidence bounds are plotted for the $\delta^{18}O_{sw}$ proxies (**a–d**), 95% confidence bounds for the Florida Straits' transport reconstruction (**e**), and 100-year filtered data alone (thick lines) is plotted for the other records (**f–l**). Light green box in both panels highlights the Little Ice Age (1450–1850 C. E.) and blue and red dashed lines indicate the mean of the respective intervals. Previously published $\delta^{18}O_{sw}$ records have all been reprocessed using the same calibration and uncertainty calculations as for the Garrison Basin records. All proxy locations are displayed as circles (sedimentary), triangles (speleothems), dashed boxes (tree-rings) or stars (poleward heat transport) in Fig. 1

in the northern Atlantic Ocean bolsters our confidence in implicating a weakened Gulf Stream and reduced surface-ocean circulation as an important dynamical process during the LIA.

In comparing available reconstructions of precipitation during the LIA with our correlation map (Fig. 1), we find remarkable agreement with the proxy record: tree-ring-based PDSI reconstructions in southern North America[42], and stalagmites from southern Mexico[43] and Peru[44] capture a wetter LIA (Fig. 4g–i) compared to modern times whereas a lake record from southern Ghana[45], titanium percent in Cariaco Basin sediments[46], and reconstructed PDSI in the southeast U. S. indicate dry LIA conditions (Fig. 4j–l). Additional proxy records appear to corroborate this observation as well (brown and green squares in Fig. 1; Supplementary Table 1). These mean state changes during the LIA all appear to be coeval with an anomalously fresher northern Atlantic Ocean, indicative of weakened Gulf Stream strength and reduced surface-ocean circulation.

Coupled general circulation model (GCM) simulations show that weakened AMOC and associated reduced heat and salt transport[47] can arise from sustained sea-ice-ocean feedbacks[7,48], triggered either by external forcing[49,50] or through internal ocean-atmosphere interactions[6,51], both of which can result in a southward displacement of the Atlantic intertropical convergence zone (ITCZ)[49,52,53]. This state of weakened AMOC, observed in millennial-scale and glacial paleo-studies[54,55], with cool and fresh north Atlantic anomalies and a southward ITCZ, can induce increased rainfall over the southwest US via atmospheric teleconnections associated with the North Atlantic subtropical high overlying the gyre[56–58]. Despite this southward shift, positive SSS

anomalies can occur in the tropical Atlantic (and negative anomalies in the northern Atlantic) due to reduced freshwater input resulting from decreased rainfall in the Amazon and West African regions[47]. Eventually, the tropical positive salinity anomaly in the southern Atlantic propagates northward, thereby strengthening meridional oceanic transport[49] and providing the delayed negative feedback.

We investigated this centennial-scale link between salinity and precipitation variability in a transient simulation of the last millennium using the fully-coupled MPI-ESM-P model with prescribed external forcing[7,59]. We perform similar correlation analysis on the model output between GOM SSS in the model and (1) global oceanic SSS and (2) continental precipitation simulated by the model, this time on centennial time-scales (Fig. 5). The resultant correlation map contains features that strongly resemble those from the multidecadal correlation analysis performed using reanalysis datasets (Fig. 1). Though the length of the instrumental record limits us from directly analyzing centennial-scale correlations, there is theoretical and modeling evidence to implicate similar ocean-atmosphere processes on multidecadal and centennial timescales[6,47,49]. Both model and observational analyses reveal a dipolar structure in Atlantic Ocean SSS that is consistent with the LIA proxies and thereby supports our hypothesis linking meridional salt transport and tropical rainfall[49]. Both analyses also display similarities in continental precipitation patterns over western Africa, northern South America, and the southwestern United States, which are also consistent with the LIA hydroclimate proxies. However, there are significant differences between the correlation patterns as well. Most notably, rainfall

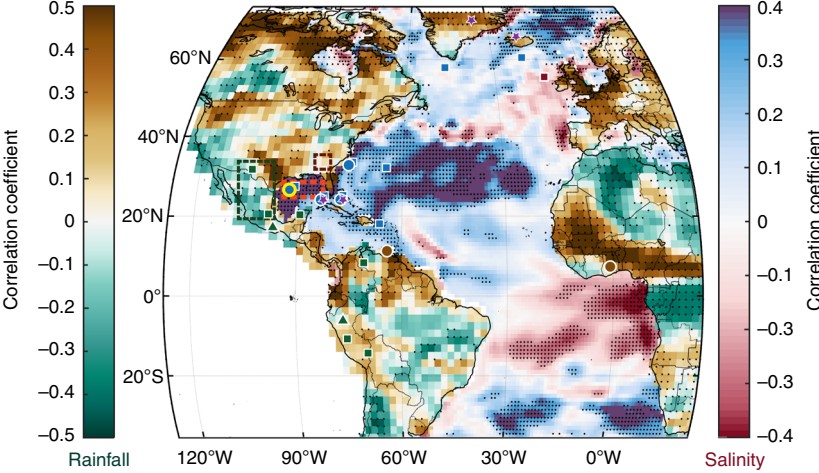

**Fig. 5** Simulated correlations between sea-surface salinity and rainfall over last millennium. Correlation map between northern Gulf of Mexico SSS (dashed red box) and global oceanic SSS (red-blue scale) as well as continental precipitation (brown-green scale) from the MPI-ESM transient simulation of the last millennium along with locations of proxy records used in the study. Proxy markers are filled as in Fig. 1. Correlations were performed with 50–150 year bandpass filters to isolate centennial scale variability, where black stippling indicates significance at the 5% confidence level

correlations over North America, Europe, and western South America, and salinity patterns offshore Europe are in disagreement between the model output and the reanalysis data-based correlations. Future work is required to investigate whether these occur due to biases in the model, insufficient observations, or whether these might indicate significantly different responses on multidecadal versus centennial timescales. Regardless of these discrepancies, the broad agreement between the analyses supports similar ocean-atmosphere processes on multidecadal-to-centennial timescales[6,47,49] and provides additional evidence of a robust century-scale link between circulation changes in the Atlantic basin and precipitation in the adjacent continents.

The transient simulation indicates that a weaker gyre, increased sea-ice cover, and reduced interhemispheric heat transport causes the ITCZ to shift southward and produces anomalous rainfall over the Americas[55,58,60]. During the onset of the LIA in the transient simulation, fresh anomalies are observed in the Arctic that serve to weaken the subpolar gyre abruptly[7]. Sensitivity experiments with various forcings and slight changes in the initial conditions before the onset of the LIA indicate that though volcanic forcing can aid in inducing such conditions[50], internal ocean dynamics can produce a weakened subpolar gyre[7,8,61,62]. Such changes are also associated with a slowdown of the western subtropical gyre[7], in agreement with reconstructed evidence of a weaker Gulf Stream and a weaker subpolar gyre during the LIA[9,11,41,63,64].

Interestingly, sensitivity experiments with the MPI-ESM last millennium model simulations indicate that anomalous changes in gyre strength and its effect on SSS and surface-circulation need not be coupled with changes in AMOC strength[7]. These simulations indicate that changes in the surface-ocean circulation need not be driven solely by changes in the AMOC system. For example, anomalous behavior of the subpolar gyre and its associated changes in surface circulation might be indicative more of a response to wind-driven, ocean-atmosphere interactions and changes in sea-ice cover[7,48] rather than being controlled only by changes in deepwater formation (and the AMOC system as a whole) on these timescales. More quantitative and highly resolved proxies of North Atlantic deepwater formation are required to refute or confirm whether this was the case during the LIA and refine our understanding of whether a 20th-century slowdown in AMOC is unprecedented in context of the last millennium[13], a matter of significant debate in the literature[65–67]. However,

regardless of the specific physical mechanism concerning the onset of the LIA[7,34,50,68] and whether AMOC changes were linked with circulation changes in the surface ocean, we hypothesize that the reported oscillatory feedback on centennial-time scales involving the surface-circulation in the Atlantic Ocean and Western Hemisphere hydroclimate played an important role in last millennium climate variability and perhaps, over the late Holocene.

The new, triplicated Garrison Basin paleoclimate records serve to extend the historical perspective of sea-surface conditions over the last 4,400 years and provide important constraints on low-latitude Atlantic Ocean surface circulation variability on centennial timescales. Robust uncertainty analysis of these reconstructions provide confidence in documenting substantial centennial-scale variability in SST and $\delta^{18}O_{sw}$ whose magnitude model simulations should be tested against. Our proxy synthesis reaffirms the role of weakened Atlantic Ocean surface circulation during the onset of the LIA and also demonstrates a centennial-scale linkage with Western Hemisphere rainfall variability, offering new insights into late Holocene climate change.

## Methods

**Multicore collection and sample processing**. Multicores 2010-GB2-MCA, 2010-GB2-MCB, and 2010-GB2-MCC (hereafter MCA, MCB, and MCC) were collected in the Garrison Basin, northern Gulf of Mexico (26° 40.19′N,93° 55.22′W) during the summer of 2010 aboard the R/V *Cape Hatteras* at a water depth of 1776 m. Cores MCA (core depth = 60 cm) and MCC (core depth = 58 cm) were both extruded and sectioned every 0.5 cm over their entire depths. MCB (core depth = 57 cm) was sectioned at 0.25 cm and analyzed up to 18 cm to obtain higher-resolution sampling over the last millennium. After extrusion, the mud samples from the three cores were disaggregated and gently sieved through a 63 μm mesh size sieve using ultra-pure water, and subsequently dried.

**Age-depth modeling and uncertainties**. The three cores were independently dated using AMS radiocarbon measurements ($n = 12$: MCA = 6, MCB = 3, MCC = 3: no age inversions; Supplementary Figure 1) on upper-ocean planktic foraminiferal species (primarily *Globigerinoides sacculifer*, *Globigerinoides ruber* (white), and *Globigerinoides conglobatus*). Each multicore contained intact and excellently preserved sediment-water interfaces and the tops of all three cores contained bomb radiocarbon, confirming that the core-tops contained the most recently deposited sediments and were not significantly bioturbated. Resultant sedimentation rates of ~14–15 cm kyr$^{-1}$ allowed for ~30-year sampling resolution (for MCA and MCC; MCB was sectioned every 0.25 cm for ~15-year resolution over the last millennium).

The radiocarbon dates for each multicore were individually calibrated into calendar years using the R code BACON[17]. This code uses a Bayesian Monte Carlo Markov Chain (MCMC) methodology to produce numerous age-depth models

with inherent reservoir-age-based and past-[14]C-variability-based uncertainty. In this process, we applied the Marine13 curve[69] to calibrate for marine reservoir ages and used a correction ($\Delta R$) of -32 ± 25 years, based on coral radiocarbon ages from the nearby (~100 km) Flower Garden Banks Sanctuary[70]. With post-bomb core-top ages and the BACON output, we built 5,000 age-depth models for each of the three cores based on a monotonic Monte Carlo procedure that included the full extent of their calendar age probability distributions (Supplementary Figure 1).

To investigate the robustness of the individual age-depth models against the combined ages of all three cores, we calibrated and built a set of 5,000 age-depth models using all of our ages ($n = 12$; black line in Supplementary Figure 1) based on the aforementioned procedure. This exercise showed that, within uncertainty, the three, individual age-depth models were indistinguishable from the combined age model and validated the assumption that all three cores from the same multicore cast also had the same sedimentation rates. We used the age-depth model and inherent uncertainties containing all the ages (most informative) to subsequently transform stratigraphic depth into calibrated ages.

**Stable isotopic and Mg/Ca analyses.** 70–100 individual specimens of the white variety of planktic foraminifer *Globigerinoides ruber* (*G. ruber* (W)) were picked from the 250 to 300 µm fraction in each sample for geochemical analysis. This narrow fraction window (50 µm) minimizes size-based ontogenetic effects[71]. The geochemical variability of different morphotypes of *G. ruber* (W) was found to be insignificant in the northern Gulf of Mexico, based on sediment trap, core-top, and downcore samples[16]. The picked tests were visibly inspected for manganese and organic-based contaminants prior to analysis, gently crushed, homogenized, and separated into two aliquots for stable isotopic and trace metal measurements.

$\delta^{18}O$ and $\delta^{13}C$ measurements ($n = 408$) were carried out using one of the aforementioned aliquots (which usually contained enough material for replicate analysis) after ultrasonicating briefly in methanol and analyzed on a Kiel IV Carbonate Device coupled to a Thermo Scientific MAT 253 Isotope Ratio Mass Spectrometer, housed at the Analytical Laboratory for Paleoclimate Studies (ALPS), University of Texas at Austin. The resulting stable isotope values are all reported in permil relative to the Vienna Pee Dee Belemnite standard notation (‰, VPDB). Two external standards, Estremoz ($n = 46$) and NBS-19 ($n = 50$) were routinely analyzed to monitor the accuracy of the measurements. The 1σ analytical precision of standard analysis during this study was ±0.03‰ for carbon and ±0.06‰ for oxygen, consistent with the long-term precision of the instrumental setup (0.06‰ for $\delta^{13}C$ and 0.08‰ for $\delta^{18}O$). Sampling uncertainty was determined to be 0.1‰ for carbon and oxygen, based on inter-sample replicate isotopic measurements ($n = 100$) of different sets of foraminifera from the same depth, consistent with forward modeling experiments in the northern Gulf of Mexico[16,18], and incorporated into our overall uncertainty analysis.

Trace element analysis ($n = 350$) was performed following routine foraminiferal cleaning protocols[72] and measured using a Perkin-Elmer Optima 4300 dual view inductively coupled plasma-optical emission spectrophotometer[73], also housed at the ALPS. All Mg/Ca values are reported in mmol/mol. An internal gravimetric standard ($n = 90$ per run; Mg/Ca = 3.2 mmol/mol) was run after every sample to monitor accuracy, precision and was used to correct raw Mg/Ca data for drift[74]. A series of gravimetric standards and an external standard (ECRM 752−1) were also analyzed with every batch of samples to assess the accuracy and precision of the instrument, whose accuracy was confirmed by an interlaboratory calibration[75,76]. We found no relationships between Al/Ca and Mg/Ca, nor Fe/Ca and Mg/Ca, and all samples in this study contained less than detectable Mn/Ca. The average 1σ analytical precision was ± 0.15% (0.010 mmol/mol) for samples in this study, based on repeated analysis of standards and 20% sample replicates. The sampling uncertainty, inferred from replicate measurements of different sets of *G. ruber* (W) tests from the same sample depth ($n = 50$), was found to be ± 3% (0.10 mmol/mol). These two sources of uncertainty were incorporated into our error propagation analysis.

**Seawater $\delta^{18}O$ and SST reconstructions and error propagation.** The paired Mg/Ca and $\delta^{18}O$ measurements of *G. ruber* (W) were used to reconstruct sea-surface temperatures (SST) and seawater $\delta^{18}O$ ($\delta^{18}O_{sw}$) using published calibration equations in a bootstrap Monte Carlo framework[16,18,19]. The algorithm employed to do so, recently published as PSU Solver[19], accounts for inherent analytical error, sampling uncertainty obtained from replicates and forward modeling, age uncertainty as modeled above, and calibration uncertainties.

At each sampling depth, we allowed Mg/Ca and $\delta^{18}O$ values to vary between their associated combined analytical and sampling uncertainty[18] to produce probability distributions around the measured value ($n = 5000$). Each sampling depth was also associated with a probability distribution of calibrated calendar age ($n = 5000$). Supplementary Figure 2 displays the uncertainty cloud for Mg/Ca measurements on 2010-GB2-MCA as an example. Using these uncertainty distributions, we acquired simultaneous solutions for SST and $\delta^{18}O_{sw}$ at each depth by inverting the following relationship equations:

1. $\ln(Mg/Ca) = 0.084 * T + 0.051 * S − 2.54$ [77]
2. $T = 16.5 − 4.8(\delta^{18}O_c - \delta^{18}O_{sw} + 0.27)$ (Low-light equation) [78];
3. $\delta^{18}O_{sw} = 0.55 S − 18.98$ (North Atlantic) [79];

where $T$ is the temperature of the water mass, $S$ is the salinity, and $\delta^{18}O_c$ is the $\delta^{18}O$ of foraminiferal carbonate. The 0.27‰ term is subtracted from the $\delta^{18}O_{sw}$ term to convert from the VPDB scale to the Vienna Standard Mean Ocean Water (VSMOW) scale. All $\delta^{18}O_{sw}$ values are reported in permil relative to this scale (‰, VSMOW).

This methodology accounts for the influence of salinity on foraminiferal Mg/Ca[80–82] by using a newly published equation[77] that is built using a collation of culture-based relationships[83–86]. A potential drawback of using this methodology is the assumption of stationarity in using the salinity-$\delta^{18}O_{sw}$ relationship back in time[87]. However, this assumption should be valid considering the lack of abrupt and significantly large changes in the late Holocene. This assumption is further validated by an isotopic modeling study that shows little-to-no salinity bias in the salinity-$\delta^{18}O_{sw}$ relationship in the Gulf of Mexico during the Last Glacial Maximum and Heinrich Event 1, two time periods with significant changes compared to the late Holocene[88]. Furthermore, the structure of the resultant $\delta^{18}O_{sw}$ and SST reconstruction at the Garrison Basin is unchanged even if we use the conventional methodology of a straightforward Mg/Ca-SST relationship[89], though the absolute magnitudes differ.

For independence from the assumption of normality (which is generally not true for these uncertainty distributions due to age calibration error), after inverting for SST and $\delta^{18}O_{sw}$ from our multicore-based geochemical records, we used the median, and 32nd–68th and 5th–95th percentiles, as the corresponding value and its confidence bounds at each sample depth, whose time uncertainty was also incorporated into the propagated uncertainty envelope as delineated in the publication that details PSU Solver[19]. We also investigated a blanket bioturbation-related uncertainty[90] of ~2 cm and found no differences in the structure of the resultant time series.

**Stacking methodology.** We produced a stacked record of Mg/Ca and $\delta^{18}O$ variability from the three Garrison Basin multicores and resultant stacked records of $\delta^{18}O_{sw}$ and SST variability over the past 4400 years, maximizing accuracy in extracting climatic signals and minimizing the errors on the reconstructions (Fig. 2 in main text). The stacked record was also produced in a bootstrap Monte Carlo framework ($n = 5000$) where the median values from the overall uncertainty probability distributions of MCA, MCC, and MCB (when available) are used as the stack values. The 32–68th and 5–95th percentiles of this distribution were used as the subsequent confidence bounds (Supplementary Figure 2). This methodology 'rewards' the stack by reducing the uncertainty envelope when all three multicore distributions are closer together, and 'punishes' the stack with larger errors when they are different (Fig. 2). This ensures that the ensuing uncertainty envelopes minimize methodological and geological uncertainties where geochemical values from the same depths in different cores can be dissimilar, a common occurrence in foraminiferal-based replication studies of cores from nearby basins[16,31,91,92]. To our knowledge, this is the first study to produce paleoceanographic records from the same basin using three multicores from the same cast. No mean offsets or corrections were applied to any of the core data to produce stacked records of SST and $\delta^{18}O_{sw}$ variability.

**Comparisons with observations.** We used available measurements of $\delta^{18}O_{sw}$ close to the Garrison Basin to compare with the stacked $\delta^{18}O_{sw}$ reconstruction. These measurements were made on surface waters in the northern Gulf of Mexico taken from CTD-casts, analyzed on a Thermo Scientific Gasbench II coupled to a Thermo Scientific Delta V isotope ratio mass spectrometer housed at the ALPS. Since these measurements do not span a complete year, we also included $\delta^{18}O_{sw}$ data from the proximal Flower Garden Banks to produce a modern data point (orange) plotted in Fig. 2e.

Where overlap was available, we compared the Garrison Basin SST stack with instrumental observations from the HadISST data set[22]. To facilitate an apples-to-apples comparison, after extracting monthly observations from 1870 to 2010 from the Garrison Basin gridpoint (26.5°N,93.5°W), we divided the data set into sets of 30 years (the average sampling resolution of the stack). Next, with $n = 50$ months, we employed a bootstrap picking procedure which produced a set of 5000 distributions for the average of 50 random months selected from each 30-year set. We used $n = 50$ months to simulate 50 *G. ruber* (W) specimens that might have lived during the time period[18]; this is a conservative estimate of the homogenization procedure of the actual 70–100 foraminifera used in the study. However, the resulting medians for $n = 30$ or $n = 100$ were statistically indistinguishable. The median and 95% confidence bounds of these distributions are plotted in orange in Fig. 2h.

**Changepoint algorithm and SST-$\delta^{18}O_{sw}$ running correlation.** We used a Bayesian changepoint algorithm in MATLAB[25] to find the posterior probability of a statistically significant changepoint in the median $\delta^{18}O_{sw}$ timeseries. The Little Ice Age peak (Fig. 2e) is statistically robust even if we minimize the minimum distance between adjacent change points, or the number of changepoints to be detected.

To perform a running correlation between SST and $\delta^{18}O_{sw}$, we used the uncertainty envelopes for each of the respective time series and produced a probability distribution of 5000 running correlation values whose median and 95% confidence bounds are plotted in purple in Fig. 2e. We used a timestep of 180–200

years to investigate centennial-scale variability and to minimize high-frequency structure in the running correlations.

**Assessing the influence of bioturbation on our results**. X-ray computed tomography and visual examinations of the cores revealed minimal bioturbation where the evidence of burrowing and effects of winnowing was negligible. Despite this, due to the absence of laminated sediments, we investigated the potential effects of bioturbation on the resultant records using statistical modeling. First, we used smoothing filters larger than the size of our calibrated age error (3 cm–~150 years) and applied the TURBO2[93] algorithm to our replicated stack record. We utilized mixing depths based on a range of published observations, calculated using radionuclides[94], as the input for TURBO2. The results for the range of mixing depths matched each other, and revealed that centennial-scale variability, including the anomaly observed during the LIA, would still be preserved, despite bioturbation. We found that our changepoint algorithm identified the $\delta^{18}O_{sw}$ changepoint at the onset of the LIA regardless of the modeled signal of bioturbation. The algorithm was unable to positively identify a changepoint at the LIA only when the mixing depths we applied were unrealistically high (> 20 cm) for the Gulf of Mexico slope basins[94]. Moreover, the individual MCA/MCB/MCC multicores reproduce each other and thus provide high confidence that we are indeed reconstructing signals of climate variability. Thus, through replication and error propagation, we constrain age uncertainty and include the errors into our confidence bounds before interpreting our downcore reconstructions.

**Marine proxy screening and reprocessing**. To investigate regional SST variability over the last millennium, we compiled published mean-annual, mixed-layer SST records from the Gulf of Mexico: Pigmy Basin: PBBC–2[30], Fisk Basin: PE07-5I and (previously published) Garrison Basin: PE07-2, Dry Tortugas: W167-79GGC and Grand Bahama Bank: KNR166-2-118MC-A[32] (note: all of these studies utilized *G. ruber* (W) as proxies; see Supplementary Table 1). As detailed above for the multicores used in this study, we used the $^{14}$C ages on each of these cores and reprocessed them through BACON to produce calendar age distributions at each depth. Sampling uncertainty calculated using INFAUNAL[18] and the reported analytical uncertainty was used to produce error distributions for these datasets. Except PE07-5I and PE07-2, all other records used paired Mg/Ca-$\delta^{18}$O analyses so we could use the same methodological uncertainty procedures as above to calculate SST and its uncertainty envelope (Fig. 3a). For PE07-5I and PE07-2, we used the straightforward equation of Mg/Ca = $0.449e^{0.09*SST}$ [89] along with the low-light equation[78] in an iterative bootstrap framework ($n = 5000$) to calculate SST and its associated uncertainty (as there is no published $\delta^{18}$O record from the core).

To investigate past $\delta^{18}O_{sw}$ variability across the Atlantic (Figs. 1, 4, and 5), we chose available records of mean-annual surface water $\delta^{18}O_{sw}$ spanning the last millennium with at least four continuous points across 1450–1850 C. E. and at least two data points from 1850-2010. These criteria were utilized in compiling Supplementary Table 1. We omitted records that did not span 1450–1850 C. E. (e.g. Hall et al. 2010's record from the Gardar Drift[95]) as we wanted to investigate the full expanse of the LIA (1450–1850 C. E.) and those records of $\delta^{18}O_{sw}$ that were reconstructed using sub-surface proxies (such as *G. inflata*[40]) or seasonally sensitive proxies (such as *N. pachyderma*[96–98]) to facilitate a mean-annual surface-ocean-only comparison.

The Dry Tortugas, Great Bahama Bank, and Carolina Slopes had companion cores from nearby sites. For Figs. 3 and 4, we used the cores with higher sampling resolution for comparison. However, these companion cores were similarly reprocessed and showed similar structures to their proximal cores. For centennial-scale comparisons, all proxies shown in Figs. 3a and 4 (including precipitation/ NADA proxies), were smoothed with a 100-year lowpass filter[99]. The NADA tree ring records were initially smoothed with a 10-year lowpass filter, which was eventually filtered again This was also performed in a bootstrap Monte Carlo framework to produce confidence bounds on the smoothed time series.

**Observation-based correlation map and data analysis**. We used the gridded ORA-S4 monthly salinity data set (1958-2013) to correlate sea-surface salinity (SSS) in the northern Gulf of Mexico (red box in Figure 1; 23.5°N–28.5°N, 83.5°W–95.5°W) with global ocean SSS[26]. The data set was annually averaged, filtered with an 8-year lowpass filter to infer multidecadal variability and remove sensitivity to interannual variability, and then detrended prior to correlation. We used eight years to optimize inferring multidecadal variability and to ensure sufficient degrees of freedom for correlation analysis due to the length of the data set. The statistical significance of the resulting correlation coefficients was assessed by a one-tailed Student's *t*-test at the 1% confidence level. We estimated the effective sample size ($N_{eff}$) by dividing the number of years ($N$) by the mean characteristic time scale ($\tau$), defined as the minimum lag for which the autocorrelation falls below 0.2[100,101]. This methodology is especially useful where $N_{eff}$ tends to be low. Our significance analysis was verified using a Monte Carlo-based methodology[27] which also yielded similar results. We also performed similar correlations with Gulf of Mexico SSS and evaporation-minus-precipitation at every grid point from the ERA data set, which yielded remarkably low and localized correlations whose spatial patterns were not similar to the oceanic SSS correlations. Furthermore, the correlation between Mississippi outflow alone and the northern Gulf of Mexico

salinity was also low. Thus, we invoke oceanic current changes as the most plausible explanation for the initial correlation analysis.

Using the same methodology, we also correlated long-term (8-year) SSS in the northern Gulf of Mexico with land-based rainfall observations using the gridded GPCC data set[29] over their time of overlap (1958-2010), which is plotted in green/ brown in Fig. 1. The calculation of statistical significance for these correlations was also performed as described above.

**Last millennium simulation and data analysis**. The Max Plank Institute Earth System Model for paleo-applications (MPI-ESM-P) contains the atmosphere general circulation model ECHAM6, run at a horizontal resolution of spectral truncation T63 (1.875°) with 47 vertical levels, as well as the ocean/sea-ice model MPIOM which is run at 1.5° resolution and contains 40 vertical layers[8,59,102]. The transient simulation was run from 850 to 2005 following the Paleo-Modeling Intercomparison Project Phase 3 (PMIP3) protocols[103,104] with prescribed external forcing factors including volcanic and anthropogenic aerosols, greenhouse gases, total solar irradiance, orbital parameters, and changes in land-cover[8,59,102].

For the correlation analysis in Fig. 5, we used a 50–150 yr bandpass to isolate the variability associated with centennial timescales. Here, we extracted monthly precipitation and salinity as simulated by the model from 1000 CE to 2000 CE and then annually averaged and bandpass-filtered the data set. Statistical significance was assessed in the manner as stated above. We also performed this correlation in the same manner as we processed the observational datasets. This comparison is plotted in Supplementary Figure 4. For illustrative purposes, we have also plotted the reconstructed versus simulated SSS in the northern Gulf of Mexico in Supplementary Figure 5.

**Code availability**. The codes that have contributed to the results and analysis reported in this study are readily available upon request from the lead author (KT: kaustubh_thirumalai@brown.edu; Git: holy-kau).

**Data availability**. All isotopic and trace metal data produced from this study on the three multicores (including radiocarbon measurements) from the Garrison Basin, northern Gulf of Mexico are archived in the Paleoclimatology Dataset repository in the National Centers for Environmental Information, NOAA database (https://www.ncdc.noaa.gov/data-access/paleoclimatology-data/datasets).

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

## Acknowledgements

We thank Johann Jungclaus for aiding interpretation in the last millennium simulation and Thomas Guilderson for help with radiocarbon dates. This manuscript greatly benefitted from discussions with Deborah Khider, Timothy Shanahan, Pedro DiNezio, Benjamin Cook, Kevin Anchukaitis, Tripti Bhattacharya, Jessica Tierney, and Gianluca Marino. We thank the crew of the R/V *Cape Hatteras* and Team Paleo at UT Austin for help in retrieving the core samples. This work was supported by Grant #OCE-0902921 to T.M.Q. from the National Science Foundation. E.M.-C. acknowledges support from NOAA award NA16OAR4310177. The MPI-ESM-P simulations were conducted at the German Climate Computing Center (DKRZ). K.T. thanks the UTIG Ewing-Worzel Fellowship, the JSG Lagoe Micropaleontology Fund, the Consortium for Ocean Leadership, and the Brown Presidential Postdoctoral Fellowship for support.

## Author contributions

K.T., T.M.Q., J.N.R., and R.Z.P. conceived and designed the study. K.T., T.M.Q., Y.O., J.N.R., J.W.P., R.Z.P., and E.M.-C. together discussed and contributed to the interpretation of data and ideas involved in this work, and ultimately to the conclusions of this research. K.T., along with input from the others, performed the analyses, generated the figures, supplemental section, and wrote the manuscript for this study.
