## [Peer Review File · Nature Communications]

Reviewers' comments:

Reviewer #1 (Remarks to the Author):

Thirumalai et al present new foraminiferal Mg/Ca-d18O data for the last 4000 years from the Gulf of Mexico (GOM). In conjunction with compiling regional salinity proxy data and terrestrial hydroclimate proxies, with model simulations, they propose a link between the AMOC and western hemisphere hydroclimate during the Little Ice Age (LIA).

This was an interesting paper that builds on existing work from this region, perhaps most notably Lund and Curry 2006. I did not find the link between their salinity proxy data (nor the compilation and model results) and inferred changes in AMOC convincing and not enough discussion/interpretation was dedicated to explain alternative hypotheses for the observed trends. By itself the work is commendable and valuable and certainly worth of publication, but does not represent a major advance in our understanding of the topic.

I had several suggestions and queries that could hopefully improve the manuscript.

The authors ought to compare and reconcile their correlations and analysis with long-term observational SSS data by Friedman et al 2017 (GRL) who reveal a trend from 1896 to the modern of freshening subpolar North Atlantic and more saline tropical/subtropical Atlantic, which would appear at odds with the inference of this study for a coherent salinification of the whole North Atlantic since the LIA. Moreover, Friedman et al use these observational data to invoke links to AMO, NAO and ITCZ shifts (not AMOC) on different parts of the Atlantic. Reconciliation/discussion of this study with the finding of Friedman et al 2017 appears necessary.

The stated resolution of 15-35 years is somewhat misleading and is an artefact that they have simply sampled one multicore at 0.25cm resolution. The sedimentation rate at this site is not particularly high (14-15cm/kyr) and unless the authors can demonstrate the core has not been influenced by bioturbation (eg laminated sediment/anoxia) then signals at this core are almost certainly smoothed on centennial timescales: 1 cm = 66 years; bioturbation on length scales of 5cm would be typical for a core with this sed rate (eg Bard et al 2001 and refs therein); thus signals of a century and longer will likely be resolved, but not multidecadal or less. Evidence for this smoothing may be seen in the reduced amplitude of the signals observed at this site compared to other regional data, as well as the smooth record obtained showing (eg fig 4). Sampling at 0.25cm, in a core with a sed rate of only 15 cm/kyr, does not mean 15 year resolution is realistically being obtained and a brief comment ought to be added to inform the reader of the likely smoothing of signals shorter than a couple of centuries. My apologies if the core is not subject to bioturbation and I have simply missed mention of this somewhere.

The use of the term LIA is used loosely and when the data are examined, they do not show a simple story, as sold, of a hydroclimate/SSS change linked to weaker AMOC during the LIA, but rather an event at ~1400AD and then a shift towards modern. The timing and nature of these changes differ between different records. The LIA is typically defined as ~1350AD to 1850AD, with the colder intervals occurring from ~1500-1800AD. The LIA as a whole does not stand out in these records, but rather there is an event and other variability occurring within the LIA. The simplification to a weaker AMOC and altered hydroclimate during the LIA is therefore not compelling.

The omission of reference to the AMOC reconstruction work of Rahmstorf et al 2015 (Nat Clim Change) is surprising and ought to be addressed. This study suggests minimal change in AMOC during the LIA (if anything, stronger) and a weakening during the twentieth century. Given the inferred conclusion of Thirumalai et al for a weaker AMOC during the LIA, this should be discussed.

Care should be taken with the use of records used to infer AMOC change. Greenland d18O should not simply be interpreted as a proxy for poleward heat transport - it is more dependent on Nordic sea-ice cover (eg Li et al 2010) and the warming over the last 1-200 years rather than reflecting a stronger AMOC is simply global warming. The variability in Florida Straits reconstruction of Lund et al 2004 may be more strongly, or at least as equally, controlled by changes in the wind driven gyre circulation rather than AMOC.

There are many omissions of salinity proxy records from the North Atlantic, and thus it appears as

if the authors have cherry picked which records they are showing to simply support their hypothesis. For example, given the requisite for the authors to show a North Atlantic wide coherent salinity response, there are additional subpolar North Atlantic salinity records resolving the LIA which could be used to complement the subtropical data (which seem sparse too) and that could be added to figure 1: Richter et al 2009 (QSR), Nyland et al 2006 (G³), Hall et al 2010 (Paleoceanography), Thornalley et al., 2009 (Nature).

The authors are to be commended for their replication of the Mg/Ca-d18O data in 3 multicore subcores, which show excellent agreement. A more challenging test would of course be to replicate at a slightly different site since all subcore would be susceptible to any errors introduced from any downslope transport, local alteration of rube habitat depth or seasonality or compounding effects such as carbonate ion changes. To a large extent the similar trends seen in the regional compilation does this. I just urge the authors not to overplay the significance of replicating data (and stacking) from the same site, since it does not eliminate all errors, such as those mentioned above as well as systematic errors in the calibration. Following on from this, the errors presented for this study are unrealistically small since they rely too heavily on the idea that replication of Mg/Ca or d18Osw between replicates at the same site can counter the errors in temperature calibration; random errors contributing to this uncertainty will be reduced but not systematic proxy bias (seasonality, depth, other environmental factors controlling Mg/Ca common to the 3 subcores (eg change in growth rate, local carbonate ion).

Lund and Curry 2006 had problems interpreting their d18Osw values and the range obtained, since the local d18Osw-S relation yielded unrealistically large changes, suggesting complications with its use as a SSS proxy. Further discussion of this would be useful.

I find the final link to an AMOC cause for the changes a bit of a stretch. As is discussed, the model results can cause similar changes as observed not through an AMOC change but rather simply a change in the strength of the surface gyres. In this model the LIA is not linked to a weaker AMOC but is more strongly caused by a weakening subpolar gyre and its links to sea-ice and atmospheric circulation. Given the authors do not present a strong case for an AMOC link, and there is conflicting evidence for what the AMOC did during the LIA (stronger or weaker) I would urge a more speculative tone to the AMOC link. In my opinion it is not needed – this is an interesting paper that is drawing together lots of SSS records and trying to synthesize them with terrestrial data; the AMOC link weakens it.

Reviewer #2 (Remarks to the Author):

I am very supportive of the kind of work that is reported on in this manuscript. The authors use a variety of paleoclimatic indicators to provide insights onto the role of centennial scale changes in the Atlantic ocean on the larger climate system. Specifically, they focus on records from the Gulf of Mexico that indicate sea surface temperature (SST) and sea surface salinity (SSS) conditions that may also be related to larger scale conditions in the Atlantic. From these records they infer centennial scale variations and their larger climatic associations.

I feel that this work may eventually be suitable for publication, but the primary problem I see now is that the writing is not at all sufficiently clear to be of utility for non-specialists. My background is as a climate modeler who is very familiar with issues related to Atlantic variability and climate. However, I found the discussions of the various processes and their variations to be very confusing. I strongly suggest that the authors fundamentally rethink exactly what they are trying to communicate, and that in doing so they very critically think how they structure their manuscript.

Specific comments

1. Please provide line numbers for ease of commenting on specific sections of the text.

2. Bottom of page 2 ... I would really question the statement saying that the last millennium has "... well-known external forcing ..." . Also, if the words "well-know" in that sentence also apply to "characteristics global oceanic and terrestrial fingerprints", then I challenge what is really meant by "well-known".

3. Page 3 I am not completely convinced by aspects of the discussion of the Loop Current, Gulf of Mexico reconstructions, and Atlantic conditions. There is an assumption that warmer and saltier conditions are associated with a stronger Loop current and transport through the Florida Straits. But the authors also suggest that it is really the frequency of eddy shedding and associated transport of warm, saline water into the Gulf of Mexico that influences SST and SSS at the reconstruction sites. Thus, what is the linkage between eddy shedding and the strength of the transport from the Caribbean through the Florida Straits and into the North Atlantic? This gets at the question of the real linkage between the Gulf of Mexico reconstructions and larger scale climate.

4. Some of the detailed discussion makes this article potentially inaccessible to non-specialists. For example, at the top of page 5, there is the phrase "... at the site using picking experiments performed with INFAUNAL on the HADISST dataset." I have no idea what "picking" or "INFAUNAL" mean here.

5. Millennial scale model simulations are used to provide a perspective on some of the observed relationships. When I compare Fig. 1 and Fig. 5 the correlations over North America look very different between the instrumental records (Fig 1) and the model simulation (Fig. 5). What does this imply about how we should interpret the results?

6. With regard to the model simulations ... these are simulations of the last millennium with radiative forcing changes ... but are the relationships shown by the correlation analyses a result of the imposed radiative forcing, or simply a manifestation of the types of variability that are produced from natural interactions in the model? Further, the loose similarity in the salinity correlation patterns does not imply anything about ocean circulation changes ... such patterns could be driven by large-scale shifts in atmospheric circulation. I do not find that the model results add much ... what is the relevance of the fact that these are simulations of the last millennium rather than control simulations? How well does the model reproduce a LIA, and what are the driving factors?

Reviewer #3 (Remarks to the Author):

Review of the manuscript "Pronounced centennial-scale Atlantic Ocean climate variability correlated with Western Hemisphere hydroclimate" (NCOMMS-17-11582) written by Dr Thirumalai and colleagues submitted to Nature Communications.

This manuscript presents very interesting analysis on the linkage between centennial-scale Atlantic variability and western hemisphere hydroclimate over the last millennium using multiproxy synthesis. The Garrison Basin reconstructions of SST and SSS proxy are linked to the loop current strength and thus AMOC variations. These proxies are compared with other proxies for AMOC and rainfall. The results suggest that the AMOC was weakened during the Little Ice age and had important impact on western hemisphere precipitation at various locations. The results are further supported by a transient model simulation.

The topic on the linkage between AMOC and western hemisphere hydroclimate is important and will attract wide interests in the community and the wider field. The use of Garrison Basin reconstructions of SST and SSS proxy variations as a proxy for the loop current and AMOC

strength are novel results. The multiproxy synthesis provide important information for the teleconnection between ocean dynamics and hydroclimate. The results will influence thinking in the field. Meanwhile, some important technical aspects need to be clarified in the manuscript. I recommend the manuscript be accepted for publication in Nature Communications after some minor revision outlined in the following specific review comments.

1, The manuscript analyzed the fully coupled GCM (MPI-ESM-P) simulation for the correlation between GOM SSS and global SSS/continental precipitation on centennial time-scales. However, the key linkage between the GOM SST/SSS and the loop current strength and the AMOC strength in the fully coupled GCM (MPI-ESM-P) simulation has not been shown. It would be nice to show the low frequency time series of GOM SST, SSS, loop current strength, and AMOC strength in this coupled simulation to verify this linkage.

2, The manuscript can also compare the simulated time series of GOM SST/SSS and precipitation at various locations with the corresponding paleo records.

3, Does the coupled model also simulate a AMOC weakening during the Little Ice Age?

4, Page 10, last paragraph, the manuscript discussed the southward shift of the ITCZ in response to a weakening of the AMOC strength, and could cite previous coupled modeling studies on this topic using water hosing experiments, such as Zhang and Delworth 2005; Stouffer et al. 2006, etc.

References:

Zhang, R., & Delworth, T. L. (2005), Simulated tropical response to a substantial weakening of the Atlantic thermohaline circulation. *Journal of Climate*, 18(12), 1853-1860.

Stouffer et al. (2006), Investigating the causes of the response of the thermohaline circulation to past and future climate changes. *Journal of Climate*, 19(8), 1365-1387.

Thirumalai et al. 2017: Author Response to Reviewer Comments

Author Response to major comments by Reviewer #1:

Comments: Reviewer #1 writes that our work is commendable, valuable, and worthy of publication with the following four major points that could stand to benefit our manuscript:

1. Reconcile our study with a new, long-term SSS dataset: Our analyses and interpretation need to be reconciled with and discussed alongside a newly published, long-term (1896-2013) surface salinity dataset over the northern Atlantic Ocean from Friedman et al. (2017).
2. Bioturbation and sample resolution versus replication: Reviewer #1 cautions against over-interpretation of our replicated results and asks us to clarify our interpretation with regard to the sample resolution available from the Garrison Basin cores, reconstructed climate variability, and potential effects of bioturbation.
3. Inclusion of previously published records: There are previous studies reconstructing salinity changes that we have neglected to include in our multiproxy synthesis and as such, by pointing this out, Reviewer #1 feels that we might have “cherry-picked” our analysis to support our hypothesis.
4. Causal link to AMOC during the LIA: Reviewer #1 suggests using a more speculative tone concerning our linkage of the Garrison Basin reconstructed $\delta^{18}\text{O}_{\text{sw}}$ record to potentially altered AMOC during the LIA. Moreover, the timing of the LIA hydroclimate changes and potentially coeval changes in ocean circulation requires further discussion and explanation.

Firstly, we thank Reviewer #1 for a constructive, in-depth, and positive review of our manuscript and also for supporting publication of our study. Below, we address the major points put forth by Reviewer #1 and detail how we have incorporated her/his suggestions.

Author Reply to Point 1 (New SSS Dataset): We thank the reviewer for pointing out this new study by Friedman et al. (2017). At the time of submission of our manuscript, this paper was not yet published and as such, we did not include it in our study. We have rectified this and included the study in our discussion and references. See e.g. Lines 116-122.

Friedman et al. (2017) compile sea surface salinity (SSS) data from multiple observational sources and produce a gridded, 118-year-long dataset. They utilize empirical orthogonal functions (EOFs) and principal component analysis (PCA) to establish two major regions of long-term Atlantic SSS variability, namely, the North Atlantic (NATL) and the tropical Atlantic (TATL). In the study, they show that there is a century-long salinification trend in the TATL PC time series with an EOF pattern indicating basinwide salinification. Although their SSS dataset does not include any datapoints from the Gulf of Mexico (Fig. 1 Friedman et al., 2017), such a basinwide salinification of the northern tropical-to-subtropical Atlantic Ocean is very much in line with results presented in our study. Our newly reconstructed, replicated $\delta^{18}\text{O}_{\text{sw}}$ record from the Garrison Basin, displays a secular trend from lesser-to-higher values, also indicating a century-long trend of salinification. However, as Reviewer #1 suggests later, it is important to note that our temporal resolution and associated uncertainty (in both age and $\delta^{18}\text{O}_{\text{sw}}$ values) precludes a

quantitative comparison with the new SSS dataset. We stress upon this caveat in our revised text.

Qualitatively, the Garrison Basin record and other proxy records of $\delta^{18}\text{O}_{\text{sw}}$ shown in Figure 4 seem to indicate that this century-long salinification trend is part of a multi-centennial salinification trend following anomalous freshening during the onset of the Little Ice Age (LIA), a major finding in our study. We have updated our discussion to reflect this.

On the other hand, Friedman et al. (2017) find that the NATL PC time series seems to display a long-term freshening trend whereas it switches into a salinification trend from 1970-2013 (Fig. 3, Friedman et al., 2017). Reviewer #1 feels that this 1896-2013 trend is at odds with our inference, however, we feel that this is not the case for two primary reasons:

1. The $\delta^{18}\text{O}_{\text{sw}}$ proxy records that we utilize in our synthesis, except for three records, are situated south of 45°N , the southernmost latitude for the NATL region as delineated in Friedman et al. (2017). The three proxy records that are located in this region are from the Labrador Sea¹, the South Iceland rise², and the Feni Drift³ (newly included based on Reviewer #1's suggestion). Unfortunately, the temporal resolution, age and analytical uncertainty in these records preclude a quantitative comparison with the newly compiled SSS dataset from Friedman et al. (2017). Furthermore, even if there was a freshening trend from 1896-2013 (i.e. one century-long) in the region as purported by Friedman and others, the three paleo-records indicate a multi-centennial salinification trend from 1400CE-present, and thus, a century-long, higher-resolution, freshening trend would have to be superimposed on top of this longer trend. Thus, these records support our inference that there was a basin-wide freshening event during the LIA and a multi-centennial salinification trend into the instrumental era.
2. The standard deviation of the TATL salinity timeseries is almost an order of magnitude larger than the NATL timeseries (Fig. 1b, Friedman et al., 2017) and hence there is a lower probability of proxies being able to capture and resolve potential trends in the NATL domain compared to the TATL domain.

Despite not being able to perform quantitative comparisons with these new SSS data, the Friedman et al. (2017) study is very useful for our manuscript and supports several points of note. Namely, the Friedman et al. (2017) study confirms that:

1. Meridional advection of water masses is a major physical mechanism that controls salinity variability on multidecadal timescales in the Atlantic Ocean. (e.g. TATL leads NATL by about a decade similar to Krebs and Timmermann, 2007 etc.)
2. Intensification of Atlantic Ocean circulation is associated with an increase of warm and saline surface waters into the subpolar region (and vice versa).
3. Though Friedman et al. (2017) invoke the North Atlantic Oscillation (NAO), the Atlantic Multidecadal Oscillation (AMO), and shifts in the intertropical convergence zone (ITCZ) to explain various components of their datasets, they do not rule out that the Atlantic Meridional Overturning Circulation (AMOC) might have a role in influencing the AMO or vice versa⁴⁻⁶. Our study does not preclude the influences of the NAO or AMO on the AMOC or vice versa. We refrain from mentioning those modes of variability as they operate on timescales shorter than available temporal resolution of our reconstruction and synthesis. However, we have included a line to highlight the importance of sub-centennial processes and the role they might play in explaining SSS variability. Lines 230-233.

Author Reply to Point 2 (Bioturbation and resolution): We include a section in the supplementary discussion about sample resolution and bioturbation. As stated there, we used X-Ray computed tomography and visual examination to investigate the effect of bioturbation on our cores. These inspections seemed to indicate a minimal influence of bioturbation (e.g. negligible burrow sizes etc.) Regardless, as the Reviewer suggests, since the Garrison Basin is not anoxic, it does not contain laminated sediments. Therefore, we used statistical modeling to investigate the effect of bioturbation and how it would smooth climatic signals in a foraminiferal record based on our reconstructed results. We used the TURBO2 code, a MATLAB algorithm^{7,8} that simulates bioturbation, based on input mixing depths. We used a range of published mixing depths based on radionuclide measurements in the Gulf of Mexico⁹ as input. Essentially, we can summarize our results as follows: despite the range of mixing depths applied, simulated signals of bioturbation did not result in the removal of centennial-scale variability and more importantly, each simulation resulted in the positive identification of a changepoint in $\delta^{18}\text{O}_{\text{sw}}$ records during the onset of the LIA, a central point of our study.

We have reworded our manuscript and supplementary materials to echo the Reviewer's concern regarding over-interpretation of our replicated results and associated errors. We ensure to caution the readers that though we have accounted for many forms of uncertainty, structural uncertainty in interpretation and reconstruction is still prevalent.

Author Reply to Point 3 (Previously published records): We thank the reviewer for making us aware of additional studies that reconstruct $\delta^{18}\text{O}_{\text{sw}}$ in the north Atlantic and have updated Figs. 1 and 5 to include these studies, namely, the record from the Feni Drift³ (Richter et al., 2009) and the record from the South Iceland Rise² (Thornalley et al., 2009). However, we are unable to utilize the other two records suggested by the reviewer i.e. the record from the eastern Norwegian Sea¹⁰ (Nyland et al., 2006) and the record from the Gardar Drift¹¹ (Hall et al. 2010) for the following reasons: The eastern Norwegian Sea $\delta^{18}\text{O}_{\text{sw}}$ record, comprised of measurements on *N. pachyderma* specimens, potentially record conditions that are seasonal¹⁰ and conditions that are at or below the base of the mixed layer¹²⁻¹⁴, and thus might not be a strict proxy for mean-annual SSS (additional note: the radiocarbon ages for this record are unpublished/not archived to the best of our knowledge and thus, we could not reprocesses this dataset similar to other records); the Gardar Drift record¹¹ from Hall et al. 2010, does not span the full length of the LIA (1450-1850 C.E. as defined by the IPCC, and in our study), thereby not passing our criteria for proxy screening. To make our selections clearer, we have revised our previous supplementary text on proxy screening to include the following:

*“To investigate past $\delta^{18}\text{O}_{\text{sw}}$ variability across the Atlantic (Figs. 1, 4, and 5), we chose available records of mean-annual surface water $\delta^{18}\text{O}_{\text{sw}}$ spanning the last millennium with at least 4 continuous points across 1450-1850 C.E., and at least two data points from 1850-2010. These criteria were utilized in compiling Table S1. We omitted records that did not span 1450-1850 C.E. (e.g. Hall et al. 2010's record from the Gardar Drift¹¹) as we wanted to investigate the full expanse of the LIA (1450-1850 C.E.) and those records of $\delta^{18}\text{O}_{\text{sw}}$ that were reconstructed using sub-surface proxies (such as *G. inflata*¹⁵) or seasonally-sensitive proxies (such as *N. pachyderma*¹²⁻¹⁴) to facilitate a mean-annual, surface-ocean-only comparison.”*

Author Reply to Point 4 (Link to AMOC): We agree with the Reviewer that there is conflicting evidence regarding whether AMOC variability was higher/lower during the Little Ice Age. Records of sea-surface salinity alone, such as the new Garrison Basin reconstructions and those compiled in our synthesis, cannot comprehensively constrain changes in the AMOC system.

Regardless, they can provide a perspective on changes in surface-ocean circulation variability¹⁶⁻¹⁸, that may or may not be tied to changes in the AMOC system. Moreover, as we demonstrate, such a synthesis of SSS can be correlated to proxy reconstructions of rainfall/atmospheric variability. Although the development, refinement, and production of future records that can directly track processes related to deepwater formation, for example sortable silt¹⁹ or radiogenic tracers²⁰, will provide greater constraints on changes in AMOC, we feel that $\delta^{18}\text{O}_{\text{sw}}$ records are currently underutilized in advancing knowledge on surface-ocean circulation and advection.

More pertinently perhaps, we also agree with the reviewer that sensitivity experiments performed on the MPI-ESM model indicate that anomalous changes in subpolar gyre strength (and resultant changes in Atlantic surface-ocean circulation) need not be tied to direct changes in the AMOC system as a whole. We incorporate the reviewer's suggestions by revising our Abstract, Introduction, and Discussion to take a more speculative role for the causal link with AMOC changes and deepwater formation. We have now more explicitly stated that our SSS synthesis alone cannot confirm the hypothesis that AMOC was reduced during the LIA, but rather, is consistent with this hypothesis. Furthermore, changes in the AMOC system are not needed to cause such shifts in ocean-atmosphere process that bring with it resultant spatial pattern in SSS and western Hemisphere hydroclimate during the LIA, as evinced by the model-data comparison.

Author Response to Minor Comments of Reviewer #1:

Comment: "The stated resolution of 15-35 years is somewhat misleading and is an artefact that they have simply sampled one multicore at 0.25cm resolution."

Reply: Reviewer #1 is correct in that 1 cm roughly corresponds to ~50 years and our subsampling of 0.5 cm corresponds to ~25 years (and 0.125 cm to ~13 years). As suggested, because only core MCB was subsampled at 0.25 cm and was not sampled over the length of the other cores, we have reworded the above statement to "~30 years" and readers are directed to the supplementary materials for more details.

Comment: "Evidence for smoothing may be seen in the reduced amplitude of the signals observed at this site compared to other regional data, as well as the smooth record obtained showing (eg fig 4)."

Reply: The smooth record in Figure 4 arises due to the averaging of the three different multicores and not because of a bioturbation signal. The individual records themselves are not "reduced amplitude" in nature (see Fig. 1 and Fig. S2) and are at times variable – although they do replicate. The uncertainty envelope that we calculate incorporates the disagreement between these multicores (see supplementary discussion).

Comment: "The LIA is typically defined as ~1350AD to 1850AD, with the colder intervals occurring from ~1500-1800AD. The LIA as a whole does not stand out in these records, but rather there is an event and other variability occurring within the LIA."

Reply: We utilize the canonical definition of the Little Ice Age as spanning from 1450 C.E. to 1850 C.E. as defined by the IPCC AR5. As such, our analysis centers on identifying mean-state patterns during this time period. Based on the Reviewer's suggestion, we revise our text to characterize the LIA onset as an event and that there is sub-centennial discrepancy between the inter-basin records. Regardless, all of the records display a significant difference in mean state between the modern and the LIA. See Lines 226-229.

Comment: “The errors presented for this study are unrealistically small since they rely too heavily on the idea that replication of Mg/Ca or $\delta^{18}\text{O}_{\text{sw}}$ between replicates”

Reply: Most recent studies using paired Mg/Ca- $\delta^{18}\text{O}_{\text{sw}}$ measurements report SST errors on the order of 0.5-2°C *without* replication nor the suitable treatment of error propagation (e.g. see Khider et al. 2015 for review²¹) whereas uncertainty analysis using Monte Carlo approaches can provide better constraints on error propagation as well as lessen overall uncertainty (e.g. see methods of Marino et al. 2015²²). Coupled with Monte Carlo simulations, replication of any climate record increases confidence and skill in verification-validation exercises (e.g. replication in coral records^{23,24}). We have explicitly documented the practice of using Monte Carlo approaches for error propagation (without replication) in marine sediment cores (see Thirumalai et al. 2016²⁵ and Thirumalai et al. 2013²⁶) and are confident that (1) replication reduces overall uncertainty and (2) our estimates of errors are accurate for the triplicated, stacked record wherein errors estimates on SST range from 0.6 to 1.6°C at different intervals of our downcore reconstructions (at the 5% and 95% window).

Comment: “Random errors contributing to this uncertainty will be reduced but not systematic proxy bias (seasonality, depth, other environmental factors controlling Mg/Ca common to the 3 subcores (eg change in growth rate, local carbonate ion).”

Reply: We are aware of these issues and include a note that such structural issues including these and the stationarity of the $\delta^{18}\text{O}_{\text{sw}}$ -SSS relationship may contribute to biases in the downcore reconstruction (see Lines 98-102). However, as we have stated in the paper, we have a long-running sediment trap mooring proximal to our site which constrains issues such as seasonality, depth, and other environmental factors for *G. ruber* (see Thirumalai et al. 2014²⁷). These are subject to change in past time periods, but considering that we are investigating late Holocene changes, we do not assume *a priori* that these environmental factors systematically bias our records (as opposed to say, glacial-interglacial changes).

Comment: “The variability in Florida Straits reconstruction of Lund et al 2004 may be strongly, or equally, controlled by changes in the wind driven gyre circulation rather than AMOC.”

Reply: We have incorporated this caveat into our discussion.

Comment: “Care should be taken with the use of records used to infer AMOC change. Greenland $\delta^{18}\text{O}$ should not simply be interpreted as a proxy for poleward heat transport - it is more dependent on Nordic sea-ice cover (eg Li et al 2010)”

Reply: We have added a caveat that addresses this in our discussion.

Comment: “The omission of reference to the AMOC reconstruction work of Rahmstorf et al 2015 (Nat Clim Change) is surprising and ought to be addressed. This study suggests minimal change in AMOC during the LIA (if anything, stronger) and a weakening during the twentieth century.”

Reply: We have referenced Rahmstorf et al. 2015 in our study and include discussion on the topic of anomalous AMOC changes in the 20th century.

Comment: “this is an interesting paper that is drawing together lots of SSS records and trying to synthesize them with terrestrial data”

Reply: We thank the Reviewer again for their insights, concerns, and criticisms.

References

1. Moffa-Sanchez, P., Hall, I. R., Barker, S.,

Thornalley, D. J. R. & Yashayaev, I. Surface changes in the eastern Labrador Sea around the

- onset of the Little Ice Age. *Paleoceanography* **29**, 160–175 (2014).
2. Thornalley, D. J. R., Elderfield, H. & McCave, I. N. Holocene oscillations in temperature and salinity of the surface subpolar North Atlantic. *Nature* **457**, 711–714 (2009).
 3. Richter, T. O., Peeters, F. J. C. & van Weering, T. C. E. Late Holocene (0–2.4kaBP) surface water temperature and salinity variability, Feni Drift, NE Atlantic Ocean. *Quat. Sci. Rev.* **28**, 1941–1955 (2009).
 4. Clement, A. C. *et al.* The Atlantic Multidecadal Oscillation without a role for ocean circulation. *Science* **350**, 320–324 (2015).
 5. Clement, A. C. *et al.* Response to Comment on ‘The Atlantic Multidecadal Oscillation without a role for ocean circulation’. *Science* **352**, 1527–1527 (2016).
 6. Zhang, R., Sutton, R. T., Danabasoglu, G. & Delworth, T. L. Comment on ‘The Atlantic Multidecadal Oscillation without a role for ocean circulation’. *Science* 1–4 (2016). doi:10.1126/science.aaf1660
 7. Trauth, M. H. Noise removal from duplicate paleoceanographic time-series: The use of adaptive filtering techniques. *Math. Geol.* **30**, 557–574 (1998).
 8. Trauth, M. H. TURBO2: A MATLAB simulation to study the effects of bioturbation on paleoceanographic time series. *Computers & Geosciences* **61**, 1–10 (2013).
 9. Yeager, K. M., Santschi, P. H. & Rowe, G. T. Sediment accumulation and radionuclide inventories (^{239,240}Pu, ²¹⁰Pb and ²³⁴Th) in the northern Gulf of Mexico, as influenced by organic matter and macrofaunal density. *Mar. Chem.* **91**, 1–14 (2004).
 10. Nyland, B. F., Jansen, E., Elderfield, H. & Andersson, C. Neogloboquadrina pachyderma(dex. and sin.) Mg/Ca and $\delta^{18}\text{O}$ records from the Norwegian Sea. *Geochem. Geophys. Geosyst.* **7**, Q10P17 (2006).
 11. Hall, I. R., Boessenkool, K. P., Barker, S., McCave, I. N. & Elderfield, H. Surface and deep ocean coupling in the subpolar North Atlantic during the last 230 years. *Paleoceanography* **25**, 833–9 (2010).
 12. Kozdon, R., Eisenhauer, A., Weinelt, M., Meland, M. Y. & Nürnberg, D. Reassessing Mg/Ca temperature calibrations of Neogloboquadrina pachyderma(sinistral) using paired $\delta^{44}\text{Ca}/^{40}\text{Ca}$ and Mg/Ca measurements. *Geochem. Geophys. Geosyst.* **10**, n/a–n/a (2009).
 13. Vázquez Riveiros, N. *et al.* Mg/Ca thermometry in planktic foraminifera: Improving paleotemperature estimations for *G. bulloides* and *N. pachyderma* left. *Geochem. Geophys. Geosyst.* **17**, 1249–1264 (2016).
 14. Jonkers, L., Brummer, G.-J. A., Peeters, F. J. C., van Aken, H. M. & De Jong, M. F. Seasonal stratification, shell flux, and oxygen isotope dynamics of left-coiling *N. pachyderma* and *T. quinquelobatus* in the western subpolar North Atlantic. *Paleoceanography* **25**, PA2204–13 (2010).
 15. Moffa-Sanchez, P., Born, A., Hall, I. R., Thornalley, D. J. R. & Barker, S. Solar forcing of North Atlantic surface temperature and salinity over the past millennium. *Nature Geosci.* **7**, 275–278 (2014).
 16. Thirumalai, K. & Richey, J. N. Potential for paleosalinity reconstructions to provide information about AMOC variability. *Variations* **14**, 8–12 (2016).
 17. Frankignoul, C., Deshayes, J. & Curry, R. G. The role of salinity in the decadal variability of the North Atlantic meridional overturning circulation. *Clim. Dyn.* **33**, 777–793 (2009).
 18. Polyakov, I. V., Bhatt, U. S. & Simmons, H. L. Multidecadal variability of North Atlantic temperature and salinity during the twentieth century. *J. Climate* **18**, 4562–4581 (2005).
 19. McCave, I. N., Thornalley, D. J. R. & Hall, I. R. Relation of sortable silt grain-size to deep-sea current speeds. Calibration of the ‘Mud Current Meter’. *Deep-Sea Research Part I* 1–12 (2017). doi:10.1016/j.dsr.2017.07.003
 20. Jonkers, L. *et al.* Deep circulation changes in the central South Atlantic during the past 145 kyrs reflected in a combined ²³¹Pa/²³⁰Th, Neodymium isotope and benthic $\delta^{13}\text{C}$ record. *Earth Planet. Sci. Lett.* **419**, 14–21 (2015).
 21. Khider, D., Huerta, G., Jackson, C., Stott, L. D. & Emile-Geay, J. A Bayesian, multivariate calibration for *Globigerinoides ruber* Mg/Ca. *Geochem. Geophys. Geosyst.* **16**, 1–17 (2015).
 22. Marino, G. *et al.* Bipolar seesaw control on last interglacial sea level. *Nature* **522**, 197–201 (2015).
 23. Tierney, J. E. *et al.* Tropical sea surface temperatures for the past four centuries reconstructed from coral archives. *Paleoceanography* 1–27 (2015). doi:10.1002/(ISSN)1944-9186
 24. DeLong, K. L., Quinn, T. M., Taylor, F. W., Shen, C.-C. & Lin, K. Improving coral-base paleoclimate reconstructions by replicating 350 years of coral Sr/Ca variations. *Palaeogeogr. Palaeoclim. Palaeoecol.* **373**, 6–24 (2013).
 25. Thirumalai, K., Quinn, T. M. & Marino, G. Constraining past seawater $\delta^{18}\text{O}$ and temperature records developed from foraminiferal geochemistry. *Paleoceanography* (2016). doi:10.1002/2016PA002970
 26. Thirumalai, K., Partin, J. W., Jackson, C. S. & Quinn, T. M. Statistical constraints on El Niño Southern Oscillation reconstructions using individual foraminifera: A sensitivity analysis. *Paleoceanography* **28**, 401–412 (2013).
 27. Thirumalai, K., Richey, J. N., Quinn, T. M. & Poore, R. Z. *Globigerinoides ruber* morphotypes in the Gulf of Mexico: A test of null hypothesis. *Sci. Rep.* **4**, 1–7 (2014).

Author Response to major comments by Reviewer #2:

Comment: Reviewer #2 writes that they are very supportive of the kind of work in our manuscript and detail three major points that stand to benefit our manuscript significantly:

1. Rewrite our manuscript keeping non-specialists in mind: Reviewer #2 suggests that we tone down use of terms such as INFAUNAL (Individual foraminiferal uncertainty analysis algorithm) or PSU Solver making for better readability of the manuscript.
2. Clarify our text regarding Loop Current strength and eddy shedding in the northern Gulf of Mexico: Reviewer #2 is unconvinced that NGOM SSTs can be tied to eddy shedding and greater Atlantic Ocean variability and asks us to clarify these links.
3. Address the ability of the model in simulating the Little Ice Age.

Firstly, we thank Reviewer #2 for their insightful comments and also for supporting publication of our study. Here we address their major points:

Reply to Point 1 (Terminology): We have significantly reworded our manuscript to ensure that non-specialists are not alienated while reading our manuscript. For example, we have removed terminologies such as “INFAUNAL” and instead use “bootstrap Monte Carlo simulations” (See Lines 94-96; 100-110). We have also revised our manuscript in detail so as to keep the non-specialist reader in mind, and introduce technical terminologies only in the supplemental methods.

Reply to Point 2 (Loop Current & Eddy Shedding): As the Reviewer notes, *“There is an assumption that warmer and saltier conditions are associated with a stronger Loop current and transport through the Florida Straits. But the authors also suggest that it is really the frequency of eddy shedding and associated transport of warm, saline water into the Gulf of Mexico that influences SST and SSS at the reconstruction sites. Thus, what is the linkage between eddy shedding and the strength of the transport from the Caribbean through the Florida Straits and into the North Atlantic?”*

The reviewer suggests that because of eddy shedding processes associated with the Loop Current, warmer waters that prevail in the Yucatan-to-Florida Straits penetrate into the northern Gulf of Mexico and as such, may not be indicative of the *strength* of the Loop Current. Liu et al. (2012) show that in the case of anthropogenic global warming, even though there is an increase in SSTs in the path of the Loop Current i.e. from the Yucatan into the Florida Straits, they still find that there is a net reduction in Loop Current eddy shedding and a decrease in the warm core eddy due to surface-ocean circulation such that the northern Gulf of Mexico becomes *anomalously* cooler. Gopalakrishnan et al. (2013) find that such anomalous SSTs are accompanied by SSS signatures as well. Thus, the strength of the Loop Current i.e. transport from the Caribbean through the Florida Straits are more tied to eddy shedding rather than anomalous temperatures of those waters themselves (see also Oey et al. 2005 review), as the mean annual sea-surface temperature of the waters directly in the path of the Loop Current are always warmer than mean annual SST in the northern Gulf of Mexico. We infer that *anomalously* warmer and saltier conditions prevail in the northern Gulf of Mexico BECAUSE of stronger Loop Current transport and more frequent eddy shedding into the northern Gulf of Mexico and we then tie this framework to greater Atlantic Ocean variability using correlation analyses (Fig. 1 and

5). On the reviewer's suggestion, we have clarified our text to emphasize this control of eddy shedding on SST/SSS anomalies and its link to the strength of transport (See Lines 142-154; 212-220).

Reply to Point 3 (Model Simulations): Although it is beyond the scope of this manuscript to investigate the nature of forced versus unforced variability in the model, we have reworded our discussion to comment on this aspect and reference new papers that evaluate such forcings over the last millennium (Wang et al. 2017). We also note that there is a new paper that has been published (Moreno-Chamarro et al. 2017), which we now reference, that goes into detail regarding the model's ability to reproduce a Little Ice Age. We also reference the three papers from our group that have investigated the models' ability to simulate the Little Ice Age under different forcing regimes which provide insights into the reviewer's questions (Moreno-Chamarro et al. 2015, 2016, and 2016). Essentially, fluctuations in the subpolar gyre appear to be an important process during the Little Ice Age. The model indicates that this need not be coupled to changes in the AMOC, although there are coeval changes in upper-ocean circulation, which can have impacts such as those documented by the proxies on sea-surface salinity as well as precipitation.

Reviewer #2 writes that *"I do not find that the model results add much ... what is the relevance of the fact that these are simulations of the last millennium rather than control simulations?"* We note that there are problems with all models, although, having another source of information, and the ability to investigate mechanisms is immensely useful. We justify keeping the model simulations in the manuscript as there is remarkable similarity between the correlation analysis in the model on centennial timescales and the observations on decadal timescales. As the reviewer also points out, there are important and significant differences as well: we note that these differences can be due to model biases, observational shortcomings, and/or realistic differences in the climate signal between multidecadal and centennial timescales. We have included all of these caveats into our revised discussion and incorporate the Reviewer's suggestion to make more use of the model simulations.

Author Response to Minor Comments of Reviewer #1:

Comment: "Please provide line numbers for ease of commenting on specific sections of the text."

Reply: We have added line numbers.

Comment: "If the words "well-know" in that sentence also apply to "characteristics global oceanic and terrestrial fingerprints", then I challenge what is really meant by "well-known".

Reply: We have omitted "well-known" and rephrased the sentence.

Comment: "I have no idea what "picking" or "INFAUNAL" mean here."

Reply: We have rephrased these sentences.

References

Gopalakrishnan, G., B. D. Cornuelle, I. Hoteit, D. L. Rudnick, and W. B. Owens (2013), State estimates and forecasts of the loop current in the Gulf of Mexico using the MITgcm and its adjoint, *J. Geophys. Res. Oceans*, 118(7), 3292–3314, doi:10.1002/jgrc.20239.

- Liu, Y., S.-K. Lee, B. A. Muhlring, J. T. Lamkin, and D. B. Enfield (2012), Significant reduction of the Loop Current in the 21st century and its impact on the Gulf of Mexico, *J. Geophys. Res.*, *117*(C5), C05039–8, doi:10.1029/2011JC007555.
- Moreno-Chamarro, E., D. Zanchettin, K. Lohmann, and J. H. Jungclaus (2015), Internally generated decadal cold events in the northern North Atlantic and their possible implications for the demise of the Norse settlements in Greenland, *Geophys. Res. Lett.*, *42*(3), 908–915, doi:10.1002/2014gl062741.
- Moreno-Chamarro, E., D. Zanchettin, K. Lohmann, and J. H. Jungclaus (2016a), An abrupt weakening of the subpolar gyre as trigger of Little Ice Age-type episodes, *Clim. Dyn.*, 1–18, doi:10.1007/s00382-016-3106-7.
- Moreno-Chamarro, E., D. Zanchettin, K. Lohmann, J. Luterbacher, and J. H. Jungclaus (2017), Winter amplification of the European Little Ice Age cooling by the subpolar gyre, *Sci. Rep.*, *7*(1), 339–8, doi:10.1038/s41598-017-07969-0.
- Moreno-Chamarro, E., P. Ortega, J. F. González-Rouco, and M. Montoya (2016b), Assessing reconstruction techniques of the Atlantic Ocean circulation variability during the last millennium, *Clim. Dyn.*, 1–21, doi:10.1007/s00382-016-3111-x.
- Oey, L. Y., T. Ezer, and H.-C. Lee (2005), Loop Current, rings and related circulation in the Gulf of Mexico: A review of numerical models and future challenges, *Circulation in the Gulf of Mexico*
- Wang, J., B. Yang, F. C. Ljungqvist, J. Luterbacher, T. J. Osborn, K. R. Briffa, and E. Zorita (2017), Internal and external forcing of multidecadal Atlantic climate variability over the past 1,200 years, *Nature Geosci.*, *35*, L05804–7, doi:10.1038/ngeo2962.

Thirumalai et al. 2017: Author Response to Reviewer Comments

Author Response to comments by Reviewer #3:

Reviewer #3 writes that our manuscript and analyses makes for an interesting study that will attract wide interest in the community and larger field. We thank the reviewer for their constructive comments and suggestions. They recommend publication in *Nature Communications* with the following suggestions for minor revisions. We address these below:

Comment: “It would be nice to show the low frequency time series of GOM SSS, loop current strength, and AMOC strength in this coupled simulation to verify this linkage and also compare the simulated time series of GOM SST/SSS and precipitation at various locations with the corresponding paleo records.”

Reply: The MPI-ESM model configuration that was used for the last millennium transient simulation, though it is of relatively high-resolution ($1^\circ \times 1^\circ$) compared to most state-of-the-art climate models, it is not sufficiently resolved to accurately simulate the Loop Current or associated eddy-resolving process that can be achieved with a higher-resolution, ocean-only model¹⁻³. It does however accurately simulate several features of the climate system including the large-scale circulation and associated ocean-atmosphere processes³⁻⁷. Several model-paleodata comparison studies demonstrate that it is not appropriate to compare gridpoint-to-gridpoint model output to proxy reconstructions^{8,9} without interfacing either isotope-enabled outputs¹⁰⁻¹², or using forward-modeling approaches^{13,14}, or both¹⁵ due to several factors^{8,9,16,17} including model biases, non-standardized variance, stationarity issues, and spatiotemporal uncertainty in forcing and output. It is beyond the scope of this study to accurately compare climate output at various locations in the transient simulation to the several multiproxy records synthesized, although, there are many papers in the literature that investigate the robustness of the MPI-ESM transient output as well as its deficiencies in this regard^{3,7,18-22}. We note that climate models, and specifically transient simulations, are the best available tools to investigate mechanisms of the climate system in the past and large-scale analysis of model output such as our correlation map (Fig. 5) can delineate important modes of climate variability that might explain synthesized multiproxy records. For the purposes of our discussion, the MPI-ESM simulation reveals that there is a strong linkage between surface salinity changes in the Atlantic Ocean and precipitation variability in the continental Western Hemisphere on centennial timescales, similar to that observed in the multidecadal correlation map of reanalysis observations as well as the changes observed in the proxy records between the Little Ice Age and the modern era. We have updated our discussion section to incorporate several of these points. For illustrative purposes, based on the reviewer’s suggestion, we include a figure that compares the reconstructed salinity signal in the Garrison Basin stacked record in the supplemental section.

Comment: “Does the coupled model also simulate a AMOC weakening during the Little Ice Age?”

Reply: Based on the reviewer’s suggestions, we have revised our manuscript discussion to include a paragraph that details the evolution of AMOC in the transient simulation. In short, although the model does simulate time periods of weakened AMOC over the last millennium, as detailed in Moreno-Chamarro et al. 2015, 2016, and 2016, changes in the strength of the subpolar gyre are primarily responsible for altered Atlantic Ocean surface-circulation changes during the Little Ice Age. We clarify in our revised text that our synthesis and model comparison

provide strong evidence that surface circulation was altered in the Atlantic during the LIA and only additional proxy records that directly track changes in deepwater formation can confirm that such surface-circulation changes were not coeval with changes in the AMOC system as a whole.

Comment: “Page 10, last paragraph, the manuscript discussed the southward shift of the ITCZ in response to a weakening of the AMOC strength, and could cite previous coupled modeling studies on this topic using water hosing experiments, such as Zhang and Delworth 2005; Stouffer et al. 2006, etc.”

Reply: We thank the reviewer for suggesting these references and have incorporated these hosing studies into our revised manuscript’s discussion and references.

References

1. Oey, L. Y., Ezer, T. & Lee, H.-C. Loop Current, rings and related circulation in the Gulf of Mexico: A review of numerical models and future challenges. *Circulation in the Gulf of Mexico: ...* (2005).
2. Liu, Y., Lee, S.-K., Muhling, B. A., Lamkin, J. T. & Enfield, D. B. Significant reduction of the Loop Current in the 21st century and its impact on the Gulf of Mexico. *J. Geophys. Res.* **117**, C05039–8 (2012).
3. Moreno-Chamarro, E., Ortega, P., González-Rouco, J. F. & Montoya, M. Assessing reconstruction techniques of the Atlantic Ocean circulation variability during the last millennium. *Clim. Dyn.* 1–21 (2016). doi:10.1007/s00382-016-3111-x
4. Jungclaus, J. H. et al. Characteristics of the ocean simulations in the Max Planck Institute Ocean Model (MPIOM) the ocean component of the MPI-Earth system model. *J. Adv. Model. Earth Syst.* **5**, 422–446 (2013).
5. Jungclaus, J. H., Lohmann, K. & Zanchettin, D. Enhanced 20th-century heat transfer to the Arctic simulated in the context of climate variations over the last millennium. *Clim. Past* **10**, 2201–2213 (2014).
6. Moreno-Chamarro, E., Zanchettin, D., Lohmann, K. & Jungclaus, J. H. Internally generated decadal cold events in the northern North Atlantic and their possible implications for the demise of the Norse settlements in Greenland. *Geophys. Res. Lett.* **42**, 908–915 (2015).
7. Moreno-Chamarro, E., Zanchettin, D., Lohmann, K. & Jungclaus, J. H. An abrupt weakening of the subpolar gyre as trigger of Little Ice Age-type episodes. *Clim. Dyn.* 1–18 (2016). doi:10.1007/s00382-016-3106-7
8. Ault, T. R. et al. The Continuum of Hydroclimate Variability in Western North America during the Last Millennium. *J. Climate* **26**, 5863–5878 (2013).
9. Ault, T. R., Deser, C., Newman, M. & Emile-Geay, J. Characterizing decadal to centennial variability in the equatorial Pacific during the last millennium. *Geophys. Res. Lett.* **40**, 3450–3456 (2013).
10. Dee, S. G. et al. Improved spectral comparisons of paleoclimate models and observations via proxy system modeling: Implications for multi-decadal variability. *Earth Planet. Sci. Lett.* **476**, 34–46 (2017).
11. Zhu, J. et al. Reduced ENSO variability at the LGM revealed by an isotope-enabled Earth system model. *Geophys. Res. Lett.* **44**, 6984–6992 (2017).
12. Battisti, D. S., Ding, Q. & Roe, G. H. Coherent pan-Asian climatic and isotopic response to orbital forcing of tropical insolation. *J. Geophys. Res. Atmos.* **119**, 11,997–12,020 (2014).
13. Evans, M. N. et al. A forward modeling approach to paleoclimatic interpretation of tree-ring data. *J. Geophys. Res.* **111**, G03008–13 (2006).
14. Evans, M. N., Tolwinski-Ward, S. E., Thompson, D. M. & Anchukaitis, K. J. Applications of proxy system modeling in high resolution paleoclimatology. *Quat. Sci. Rev.* **76**, 16–28 (2013).
15. Sturm, C., Zhang, Q. & Noone, D. An introduction to stable water isotopes in climate models: benefits of forward proxy modelling for paleoclimatology. *Clim. Past* **6**, 115–129 (2010).

16. Schmidt, G. A. Enhancing the relevance of palaeoclimate model/data comparisons for assessments of future climate change. *J. Quat. Sci.* **25**, 79–87 (2010).
17. Schmidt, G. A. *et al.* Using palaeo-climate comparisons to constrain future projections in CMIP5. *Clim. Past* **10**, 221–250 (2014).
18. Man, W., Zhou, T. & Jungclaus, J. H. Effects of Large Volcanic Eruptions on Global Summer Climate and East Asian Monsoon Changes during the Last Millennium: Analysis of MPI-ESM Simulations. *J. Climate* **27**, 7394–7409 (2014).
19. Zanchettin, D., Bothe, O., Müller, W., Bader, J. & Jungclaus, J. H. Different flavors of the Atlantic Multidecadal Variability. *Clim. Dyn.* **42**, 381–399 (2013).
20. Zanchettin, D., Rubino, A., Matei, D., Bothe, O. & Jungclaus, J. H. Multidecadal-to-centennial SST variability in the MPI-ESM simulation ensemble for the last millennium. *Clim. Dyn.* **40**, 1301–1318 (2012).
21. Ortega, P. *et al.* A model-tested North Atlantic Oscillation reconstruction for the past millennium. *Nature* **523**, 71–74 (2015).
22. Yan, H. *et al.* Dynamics of the intertropical convergence zone over the western Pacific during the Little Ice Age. *Nature Geosci.* **8**, 315–320 (2015).

Reviewers' comments:

Reviewer #1 (Remarks to the Author):

The authors are thanked for addressing many of the comments raised during the initial review. However, there are still two main areas which have not been satisfactorily implemented. Although the authors have taken on board some of the original comments, some of the changes have been ad-hoc insertions, rather than consistent alteration of the entire manuscript, so that the manuscript, as it stands, no longer present a clear, coherent argument. A lot of excellent work/analysis has been performed and the authors are to be commended for the rigorous science they have done in many aspects of this study, especially in utilising statistical techniques. I strongly recommend that the authors again revisit the main text, implementing the previously suggested comments (which the authors have acknowledged are valid points) to ensure consistency throughout the piece, and to present a cogent argument to the reader, which does justice to the work and effort of the authors. The two main areas for consideration are:

1. The role of AMOC, and the rationale for the study.

Although the authors have been more cautious in their wording regarding AMOC, the whole first introduction section of the paper is set up with the motivation that the purpose of this work is to reconstruct the AMOC over the late Holocene. Given the authors have conceded that factors other than AMOC may be controlling their record, and in places modified their paper accordingly, it seems incongruous that the main motivation as rationale is kept as this study being about AMOC reconstruction. Although it may be an inconvenience, I would strongly recommend that the authors work on refocussing the introduction section of the manuscript so that it is consistent with their later, correctly more cautious, interpretation. The paper can be sold on linking hydroclimate to elements of surface circulation, and some of the complexity of exactly what changes in surface circulation (eg AMOC, local circulation, regional gyre circulation etc) have occurred can already to some extent) discussed; rather than selling the study as an AMOC reconstruction, which no longer seems tenable.

On a related AMOC note: L317 - In response to the suggestion to include reference to Rahmstorf et al 2015 (ref 62; weak C20th AMOC), the authors have placed this reference very late in the manuscript, and it casually contradicts the state of the art summarized in the intro. The introduction alludes to a weaker surface Atlantic circulation (L54-60) and cites two studies (ref 9 and 10; from the Florida Straits Current and North Iceland shelf), yet ignores at this point the Rahmstorf study (ref 62) that is in direct contradiction to these two studies. Surely this is the place to mention that the results of Rahmstorf et al 2015 refutes refs 9 and 10, and the LIA may have been a period of stronger AMOC, not weaker?

2. Consistent use of onset of LIA, or LIA, and why the changes occur when they do in various records.

L135 "This event, the onset of the LIA, is also identified as a time period containing a statistically significant changepoint (grey histogram in Fig. 2g) using a Bayesian methodology considering the overall $\delta^{18}O_{sw}$ reconstruction. Thus, the LIA emerges as a unique time period"

I am glad to see in places that the authors have been more cautious and accurate in describing when their events occur eg there is an event during the early Little Ice Age. However, they have not been consistent/thorough in this and slip into calling the whole LIA anomalous (also eg L211). As stressed in the original review, their records do not show a shift that occurred throughout the LIA, but rather an event during the onset of the LIA, and then the remaining LIA is unremarkable with respect to the longer 4000 yr record. It is disingenuous of the authors to sell this as a LIA event/shift. I am concerned the authors are still trying to force the timing of their events to fit with the timing and definition of the broader LIA (eg from a European perspective of 1350-1850), when

they do not. The framework of the LIA may not be the best way to describe these data.

Describe the records accurately and summarize appropriately. Eg draw shaded boxes for the LIA (1350-1850) on Figure 2, so that it is obvious to the reader that the event being described only makes up a small portion of the LIA. This jumble of having an onset LIA event as well as then referring to the entire LIA results in a muddled story and I am not sure of the link between the 1350-1450 event seen in the author's records, versus the broader compilation which focuses on 1450-1850 (eg L223-228); there is no explicit text explaining why major SSS event in their Garrison Basin record occurs at 1450 and then recovers, whereas the other salinity changes they are discussing are averaged over a later period (1450-1850).

....And additionally

L160 "These correlations cannot be explained by changes in evaporation-minus-precipitation alone across the Atlantic Basin, and along with SST changes, are also suggestive of altered oceanic currents as the primary driver of these SSS patterns (supplementary materials)."

I am still unsure as to why the SSS pattern cannot be explained by evaporation-precipitation processes caused by coupled atmospheric circulation changes impacting the two major regions in an opposing manner. No explanation is given, and the sup. materials are referenced but no explanation is seemingly given here. I agree advection is likely the key control, but I would like the authors to explain why they rule out E-P processes.

Reviewer #2 (Remarks to the Author):

I find that the manuscript is now acceptable for publication. While the authors have not fully satisfied all of my comments, I think the manuscript will provide an important addition to the literature, and so I am comfortable recommending publication.

I would note a couple of points in passing that the authors may wish to consider before final revision:

1. In figure 1 there are actually very few continental regions with significant correlations. Some authors only color shade those points that pass some significance test. If this figure had been constructed in such a manner the area of shading over the continent would be very sparse, providing a very different impression of how well this relationship is constrained based on reconstructions.

2. The model analyses are interesting but far from completely satisfactory. Gulf of Mexico eddy shedding is invoked as a prime physical process connecting GOM conditions to the open Atlantic, but the model analyzed is not able to simulate a linkage via this mechanism due to resolution limits. This surely has an important bearing on the fidelity of the results.

Reviewer #3 (Remarks to the Author):

Review of the revised manuscript "Pronounced centennial-scale Atlantic Ocean climate variability correlated with Western Hemisphere hydroclimate" (NCOMMS-17-11582A) written by Dr Thirumalai and colleagues submitted to Nature Communications.

The revision has been improved significantly and most of my comments have been addressed. I am two remaining minor concerns:

1, In the response to my comments, it is explained that the GOM SSS is not linked to AMOC in the MPI model. I wonder then why GOM SSS is also viewed as an indicator for ocean circulation in the MPI model if it is not linked to AMOC. What causes the change in GOM SSS in the MPI model? The modeled change in the subtropical gyre is very weak (reference #56) to account for the GOM SSS change. If it is caused by the change in the Gulf Stream strength, then how much change is for the Gulf Stream strength in the model compared to the paleo observation?

2, Line 317. The 20th century AMOC slowdown and the SST based AMOC index proposed in Rahmstorf et al. 2015 are highly debated in the AMOC research community. There is insufficient observational evidence to support a finding of long term slowdown of AMOC strength over the 20th century (Rhein et al. 2013). Several recent high resolution modeling studies constrained with observational data (Jackson et al. 2016) or reconstructed freshwater fluxes (Boëning et al. 2016) suggested that the very recent AMOC slowdown since 2004 is mainly due natural variability and the anthropogenic forcing has not yet caused a significant AMOC slowdown. The revision should cite Rahmstorf et al. 2015 paper in the context of the debate.

Related References:

Boëning et al. Emerging impact of Greenland meltwater on deepwater formation in the North Atlantic Ocean, *Nature Geoscience* (2016): 523-528.

Jackson et al. Recent slowing of Atlantic overturning circulation as a recovery from earlier strengthening., *Nature Geoscience* 9 (2016): 518-522.

Rhein et al. 2013: Observations: Ocean. In: *Climate Change 2013: The Physical Science Basis. Contribution of Working Group I to the Fifth Assessment Report of the Intergovernmental Panel on Climate Change* [Stocker, T.F., D. Qin, G.-K. Plattner, M. Tignor, S.K. Allen, J. Boschung, A. Nauels, Y. Xia, V. Bex and P.M. Midgley (eds.)]. Cambridge University Press, Cambridge, United Kingdom and New York, NY, USA.

Thirumalai et al. 2017: Author Response to Second Round of Reviewer Comments

Author Response to major comments by Reviewer #1:

Reviewer #1 largely supported our revised work and commended us on “the rigorous science they have done in many aspects of this study, especially in utilising statistical techniques”. We thank the reviewer once again for their valuable comments and constructive criticism, which we largely agree with and acknowledge. She/he had two main concerns which we have now fully addressed in this second round of revision. These were:

1. The role of AMOC, and the rationale for the study

Reviewer #1 asked us to rewrite the introduction to ensure that an “AMOC reconstruction” was no longer the rationale for our study and to ensure that our overall article presented a more coherent argument rather than a disjoint one, after the first round of revision. Towards this we have made some significant revisions. We have:

1. Rewritten the abstract to emphasize the surface-circulation-hydroclimate link and omitted AMOC references
2. Rewritten the introduction paragraph to emphasize the link between surface-circulation and hydroclimate, and downplay the role of AMOC in this study, consistent with the latter discussion paragraphs. (Lines 31-42)
3. Completely omitted the introduction paragraph which discussed AMOC changes as a driver of climate variability, which had previously been used to set up our study’s rationale.
4. Emphasized that on the study-relevant timescales, surface-circulation changes in the Atlantic need not be driven purely by AMOC changes, and have also emphasized the role of the subpolar gyre and associated proxy-relevant work in our discussion.

2. Consistent use of onset of LIA, or LIA, and why the changes occur when they do in various records.

We have been more cautious in our revised manuscript about the usage of the terms “onset of the LIA” and “LIA”. Now, we explicitly include “over the entire duration of the canonically-defined LIA” in several locations to ensure that the reader is not misled regarding the LIA as only one anomaly in the Garrison Basin stack. Furthermore, we explicitly point out that the SSS reconstruction contains an “event” at the onset of the LIA and doesn’t indicate a sustained period during the LIA and call out the sub-centennial discrepancies between the other records. We also maintain that we use a “first-order” mean calculation of all values in the 1450-1850CE time period in each timeseries for the purposes of performing t-Tests and other statistical analysis. Finally, we have also removed the wording that the “LIA” is “unique” which was initially only included to describe the changepoint analyses on the $\delta^{18}\text{O}_{\text{sw}}$ record and not the SST record (although, we can now see that this is misleading and have omitted it).

Author Response to minor comments by Reviewer #1:

Comment: “Surely this is the place to mention that the results of Rahmstorf et al 2015 refutes refs 9 and 10, and the LIA may have been a period of stronger AMOC, not weaker?”

Response: We have now included the Rahmstorf et al. 2015 citation in our manuscript and refer to the debate in our introduction that it does not show evidence for a reduction in AMOC

during the LIA. This ties into our discussion section as well. We thank the reviewer for this suggestion.

Comment: “I am still unsure as to why the SSS pattern cannot be explained by evaporation-precipitation processes caused by coupled atmospheric circulation changes impacting the two major regions in an opposing manner. No explanation is given, and the sup. materials are referenced but no explanation is seemingly given here. I agree advection is likely the key control, but I would like the authors to explain why they rule out E-P processes.”

Response: We have included these lines in our supplemental text under the section “*Observation-based Correlation Map and Data Analysis*”:

“We also performed similar correlations with Gulf of Mexico SSS and evaporation-minus-precipitation at every grid point from the ERA dataset, which yielded remarkably low and localized correlations whose spatial patterns were not similar to the oceanic SSS correlations.”

For the benefit of the reviewer, we include the plot in this response:

where the figure shows the correlation between sea-surface salinity in the northern Gulf of Mexico (black box), correlated with evaporation-minus-precipitation across the domain. The EminusP dataset is from the ERA40-Interim dataset whereas the SSS is from the ORA-S4 dataset, similar to other analyses in the manuscript. All grid points with correlations with p-Values smaller than 0.1 are plotted above (note that Fig. 1 and Fig. 5 in the main text plots significance at grid points with p-values smaller than 0.01 – an order of magnitude more conservative and yet, with a higher signal-to-noise ratio).

Author Response to comments by Reviewer #2:

Reviewer #2 supports publication of our manuscript in *Nature Communications*. We thank him/her for their help and comments.

Regarding their suggestion on point-correlations and omission of non-significant grid points, we note that the model-based correlation plot (Fig. 5) includes many significant correlations over terrestrial areas and as such, refrain from plotting only significant correlations in Fig. 1 for the sake of consistency.

Author Response to comments by Reviewer #3:

We thank Reviewer #3 for their insightful comments and criticisms that have ultimately made this manuscript a better study. Reviewer #3 also supports publication of our work and has two minor concerns which we address below:

Comment: Linkages between GOM SSS and AMOC in the MPI Model:

Response: We note that we have now addressed in our manuscript why GOM SSS changed during the LIA in the model simulation whereas AMOC remained unchanged i.e. that surface-ocean circulation in the Atlantic on these timescales need not be coupled with AMOC-related processes and can have fluctuations due to internal oscillations in the regional gyre circulation. We refer the reviewer to the following publications for more detail on the MPI model and AMOC:

- *Moreno-Chamarro, E., D. Zanchettin, K. Lohmann, and J. H. Jungclaus (2015), Internally generated decadal cold events in the northern North Atlantic and their possible implications for the demise of the Norse settlements in Greenland, Geophys. Res. Lett., 42(3), 908–915, doi:10.1002/2014gl062741.*
- *Moreno-Chamarro, E., D. Zanchettin, K. Lohmann, J. Luterbacher, and J. H. Jungclaus (2017), Winter amplification of the European Little Ice Age cooling by the subpolar gyre, Sci. Rep., 7(1), 339–8, doi:10.1038/s41598-017-07969-0.*
- *Moreno-Chamarro, E., D. Zanchettin, K. Lohmann, and J. H. Jungclaus (2016), An abrupt weakening of the subpolar gyre as trigger of Little Ice Age-type episodes, Clim. Dyn., 1–18, doi:10.1007/s00382-016-3106-7.*

Comment: The revision should cite Rahmstorf et al. 2015 paper in the context of the debate.

Response: We now include the Rahmstorf et al. 2015 paper in our introduction as well as discussion in the revised manuscript and also indicate that there is ongoing debate in the literature on the topic of 20th century slowdown of AMOC and include the articles suggested by the reviewer in our discussion (Böning et al., 2016; Jackson et al., 2016).

REVIEWERS' COMMENTS:

Reviewer #1 (Remarks to the Author):

I thank the authors for implementing the suggested changes, regarding rewording the rationale and downplaying the link to AMOC. I think the revised version contains suitable discussion of the results and as such can be published. I have no further suggestions.

Reviewer #3 (Remarks to the Author):

I am satisfied with the response to my previous comments and recommend the acceptance of the manuscript.